# Understanding seed-mediated growth of gold nanoclusters at molecular level

Qiaofeng Yao [1], Xun Yuan[1], Victor Fung[2], Yong Yu [1], David Tai Leong[1], De-en Jiang [2] & Jianping Xie [1]

The continuous development of total synthesis chemistry has allowed many organic and biomolecules to be produced with known synthetic history–that is, a complete set of step reactions in their synthetic routes. Here, we extend such molecular-level precise reaction routes to nanochemistry, particularly to a seed-mediated synthesis of inorganic nanoparticles. By systematically investigating the time–dependent abundance of 35 intermediate species in total, we map out relevant step reactions in a model size growth reaction from molecularly pure $Au_{25}$ to $Au_{44}$ nanoparticles. The size growth of Au nanoparticles involves two different size–evolution processes (monotonic LaMer growth and volcano-shaped aggregative growth), which are driven by a sequential 2-electron boosting of the valence electron count of Au nanoparticles. Such fundamental findings not only provide guiding principles to produce other sizes of Au nanoparticles (e.g., $Au_{38}$), but also represent molecular-level insights on long-standing puzzles in nanochemistry, including LaMer growth, aggregative growth, and digestive ripening.

[1] Department of Chemical and Biomolecular Engineering, National University of Singapore, 4 Engineering Drive 4, Singapore 117585, Singapore. [2] Department of Chemistry, University of California, Riverside, CA 92521, USA. Qiaofeng Yao and Xun Yuan contributed equally to this work. Correspondence and requests for materials should be addressed to J.X. (email: chexiej@nus.edu.sg)

Size growth is a ubiquitous observation in synthesis, especially in seed-mediated synthesis of metals, oxides, chalcogenides, and many other nanoparticles (NPs)[1, 2]. Understanding the mechanism governing size growth is therefore central to the fundamental explorations of NPs. This understanding also largely dictates the ability of rational design and engineering of structural attributes of NPs for practical applications in diverse fields like energy conversion, environmental monitoring, biomedicine, and catalysis. Size growth of the NPs typically involves two mechanisms: LaMer growth[3] and aggregative growth[4]. In a typical LaMer growth, size growth occurs via a heterogeneous nucleation process on pre-formed small NPs (or seed NPs). On the contrary, the aggregative growth depends on agglomeration of primary NPs. Although the success of these theories has fueled the advances of synthetic chemistry of NPs in the past several decades, a lack of molecular-level understanding of growth mechanisms of NPs has become a significant bottleneck in advancing synthetic chemistries, especially in precisely customizing structural attributes of NPs for basic and applied explorations.

Thiolate-protected noble metal (Au or Ag) nanoclusters (NCs) or quantized NPs are a recent discovery of ultrasmall NPs with a core size below 2 nm, which typically possess several to a few hundred metal atoms[5–9]. They are often referred to as $[M_n(SR)_m]^q$, where $n$, $m$, and $q$ are numbers of metal atoms (M), thiolate ligands (SR), and net charge, respectively. Due to the strong quantum confinement effect in this ultrasmall size regime, metal NCs exhibit size-dependent molecular-like physicochemical properties, such as HOMO-LUMO transition[10, 11], quantized charging[12], intrinsic chirality[13–15], and strong luminescence[16–19]. Such size-dependent properties not only make metal NCs useful in catalysis[20–23], bioimaging[24, 25], energy conversion[26, 27], and sensor development[28, 29], but also provide simple and effective ways to monitor the evolution of cluster size[30–34]. Moreover, tremendous research efforts dedicated to cluster chemistry for the past two decades have enabled the production and characterization of metal NCs with atomic precision[35–38], which is not available for their larger counterparts (NPs > 3 nm or so-called plasmonic NPs). The capability of preparing metal NCs at atomic precision, together with the ease of monitoring cluster size, makes metal NCs an ideal platform to explore size growth mechanisms of nanomaterials with precision at the molecular level.

For example, we recently exemplified a homogeneous-nucleation-growth mechanism of Au NCs by a reductive formation of $[Au_{25}(SR)_{18}]^-$ from Au(I) precursors[30]. A systematic mass spectrometry analysis of the stable NC intermediates and

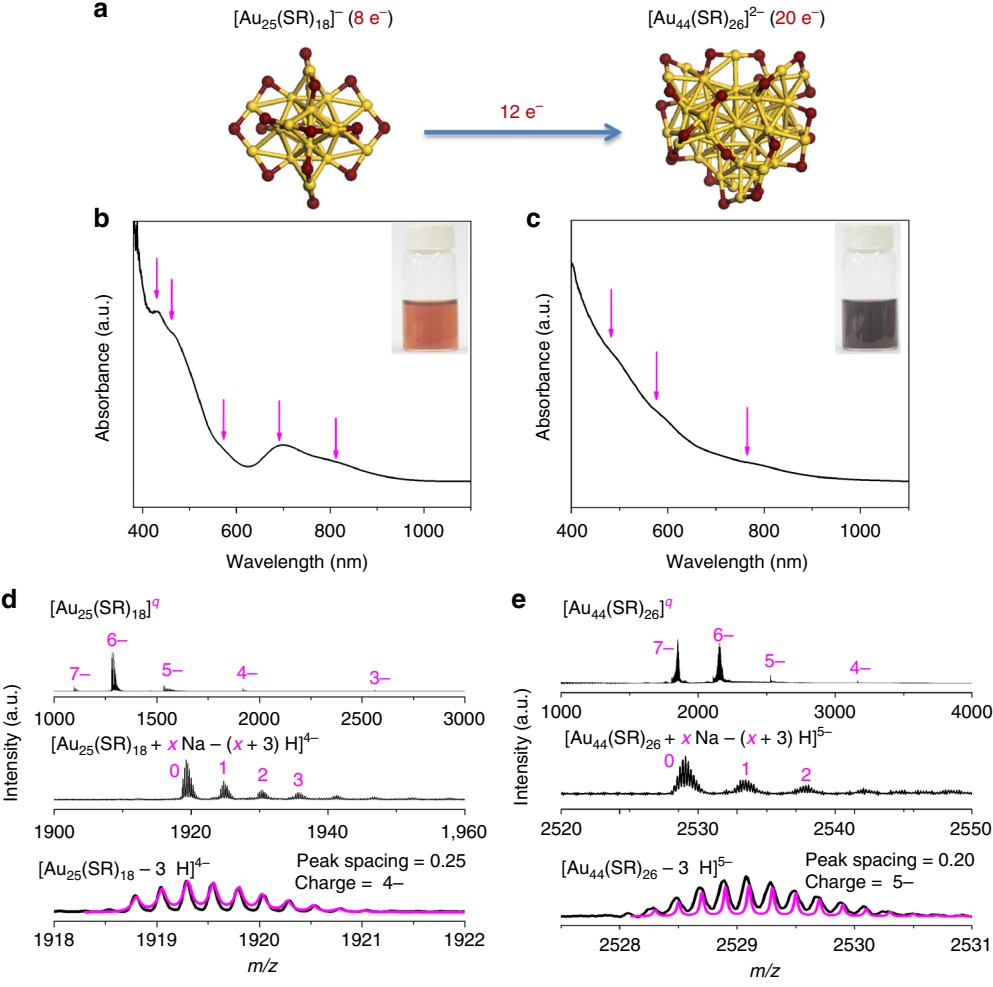

**Fig. 1** Size growth from $[Au_{25}(SR)_{18}]^-$ to $[Au_{44}(SR)_{26}]^{2-}$. **a** Schematic illustration of size growth reaction from $[Au_{25}(SR)_{18}]^-$ to $[Au_{44}(SR)_{26}]^{2-}$ (yellow, Au; purple, S). **b, c** Ultraviolet-visible absorption and **d, e** electrospray ionization mass spectrometry spectra of **b, d** $[Au_{25}(SR)_{18}]^-$ and **c, e** $[Au_{44}(SR)_{26}]^{2-}$. The crystal structures of $[Au_{25}(SR)_{18}]^-$ and $[Au_{44}(SR)_{26}]^{2-}$ are drawn according to the reported $Au_{25}S_{18}$[10] and $Au_{44}S_{26}$[15] skeletons, where all hydrocarbon tails are omitted for clarity. Insets in **b, c** are digital photos of aqueous solutions of corresponding Au nanoclusters. The magenta lines in **d, e** show simulated isotope patterns of the labeled cluster formula

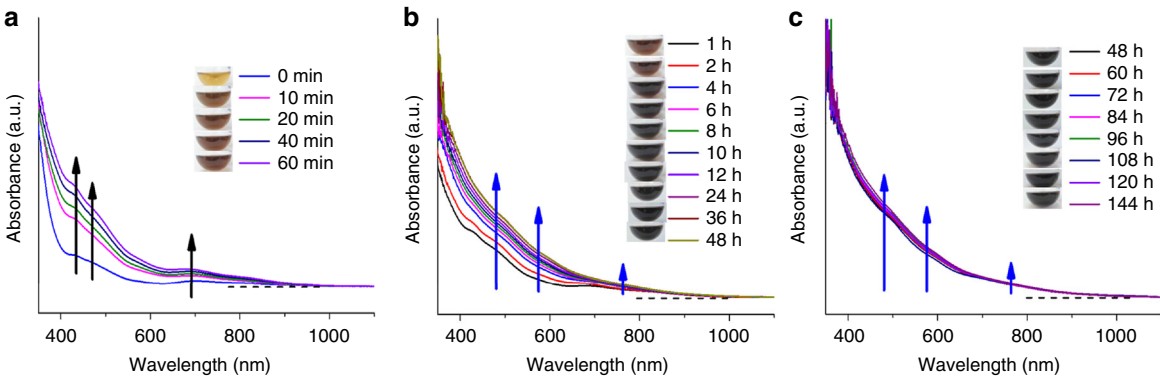

**Fig. 2** Monitoring size growth by ultraviolet-visible absorption spectroscopy. Time−course ultraviolet-visible absorption spectra of **a** pre−growth, **b** size growth, and **c** size−focusing stages from $[Au_{25}(SR)_{18}]^-$ to $[Au_{44}(SR)_{26}]^{2-}$. The insets show digital photos of reaction mixture taken at varied reaction times. The black arrows in **a** are visual guides for the absorption features of $[Au_{25}(SR)_{18}]^-$, while the blue arrows in **b**, **c** are visual guides for the absorption features of $[Au_{44}(SR)_{26}]^{2-}$

their time−dependent abundance suggests that $[Au_{25}(SR)_{18}]^-$ is formed via a two-step process. First, two electron ($e^-$) reduction converts Au(I)-SR complexes into intermediate NC species with even-numbered valence electron count ($N^* = n−m−q$). Second, a size−focusing process of these intermediate NC species finally gives rise to a stable $[Au_{25}(SR)_{18}]^-$ species. Several recent studies also successfully revealed the homogenous nucleation mechanism of Au NPs both theoretically and experimentally, suggesting the importance of chemical states and reduction pathways of Au(I) precursors in Au NP synthesis[39–42]. While these under-standings shed some light on fundamental homogeneous-nucleation-growth of Au NPs (i.e., from Au(I) precursors to stable Au NCs), a similar molecular-level understanding on the equally important heterogeneous-nucleation-growth (i.e., a small Au NC serves as a seed to grow into a larger NC species), is presently lacking. This is possibly due to the difficulty in controlling the growth of NCs at atomic precision while keeping their protecting ligands unaltered.

Herein, we present a molecular-level investigation of size growth mechanism based on a reaction from a molecularly pure $[Au_{25}(SR)_{18}]^-$ to a relatively larger $[Au_{44}(SR)_{26}]^{2-}$ species protected by identical thiolate ligands (Fig. 1a), which is made possible in a mildly reductive environment with an appropriate supply of Au(I)-SR complexes. By tracking identifiable intermediate species in the entire course of the size growth reaction, we propose a three−stage size hopping mechanism for the seed-mediated formation of $[Au_{44}(SR)_{26}]^{2-}$. These three stages are: 0) kinetically dictated accumulation of $Au_{25}$, I) $Au_{25}$-mediated size growth, and II) thermodynamically controlled size−focusing. A detailed investigation of the balanced reactions of identified intermediate species further reveals that the growth of Au NCs is driven by boosting of valence electron count either via a monotonic size growth or a volcano-shaped size−evolution pathway. It is also found that the size growth is initiated by adsorption of reductive species (e.g., carbon monoxide or CO in this study) on seed $[Au_{25}(SR)_{18}]^-$. Based on the understandings of Au NC growth, we also extend this seed-mediated growth chemistry to produce other NC intermediates, such as $[Au_{38}(SR)_{24}]^0$, of high quality.

## Results
### Seed-mediated synthesis of $[Au_{44}(SR)_{26}]^{2-}$ NCs.
The model seed cluster used in this study is water-soluble $[Au_{25}(p\text{-MBA})_{18}]^-$, where $p$-MBA denotes *para*-mercaptobenzoic acid. The synthesis of $[Au_{25}(p\text{-MBA})_{18}]^-$ was conducted according to a reported carbon monoxide (CO)-mediated reduction method[30]. The

as-obtained $[Au_{25}(p\text{-MBA})_{18}]^-$ is reddish brown in solution (inset of Fig. 1b) and exhibits characteristic absorptions at 430, 460, 575, 690, and 815 nm[30] in ultraviolet-visible (UV-vis, Fig. 1b) absorption spectrum. The electrospray ionization mass spectrometry (ESI-MS, in negative ion mode) spectrum of $[Au_{25}(p\text{-MBA})_{18}]^-$ shows 5 sets of peaks in a broad range of $m/z = 1000–4000$, which are attributed to $Au_{25}(p\text{-MBA})_{18}$ ions carrying 3-, 4-, 5-, 6-, and 7- charges, respectively (Fig. 1d, top spectrum). The detailed assignment of ESI-MS peaks was exemplified by $[Au_{25}(p\text{-MBA})_{18}–3\,H]^{4-}$ in the middle and bottom spectra of Fig. 1d. Taken the UV-vis absorption and ESI-MS spectra together, we concluded that a high quality $[Au_{25}(p\text{-MBA})_{18}]^-$ was produced by the CO-mediated reduction method.

It has been widely known that $[Au_{25}(SR)_{18}]^-$ is stable at mildly oxidative environment[43, 44]. Therefore, we hypothesized that the size growth of $[Au_{25}(SR)_{18}]^-$ should be made possible under a mildly reductive environment in the presence of Au precursors (typically Au(I)-SR complexes). Experimentally, Au(I)-SR complexes were prepared by mixing $HAuCl_4$ and $p$-MBA with a Au-to-SR ratio of 1:1 at pH 13.0, followed by the addition of purified $[Au_{25}(p\text{-MBA})_{18}]^-$ NCs. CO was then bubbled into the reaction mixture for 2 min under vigorous stirring (1000 r.p.m.). After stirring (1000 r.p.m.) for 6 days at room temperature (25 °C), high quality $[Au_{44}(p\text{-MBA})_{26}]^{2-}$ NCs were obtained as dark brown solution (Fig. 1c, inset). In sharp contrast to $[Au_{25}(p\text{-MBA})_{18}]^-$, $[Au_{44}(p\text{-MBA})_{26}]^{2-}$ exhibits only three weak shoulder peaks at 480, 575, and 765 nm in its UV-vis absorption spectrum (Fig. 1c).

ESI-MS (in negative ion mode) was then used to unambiguously determine the molecular formula of product NCs (Fig. 1e). Four sets of peaks at $m/z = 1853, 2159, 2529$, and $3162$ were identified in a broad range of $m/z = 1000–4000$ (Fig. 1e, top spectrum), corresponding to $Au_{44}(p\text{-MBA})_{26}$ ions carrying 7-, 6-, 5-, and 4- charges, respectively. All these peaks can be attributed to $[Au_{44}(p\text{-MBA})_{26}]^{2-}$, which is well supported by the perfect agreement between the simulated and experimental isotope patterns (Fig. 1e, bottom spectra). The isotope analysis together with the cleanness of ESI-MS spectrum (Fig. 1e, top spectrum) suggests a complete size conversion from $[Au_{25}(p\text{-MBA})_{18}]^-$ to $[Au_{44}(p\text{-MBA})_{26}]^{2-}$. To the best of our knowledge, this is the first observation of complete one-to-one size conversion of Au NCs/NPs capped by the same thiolate ligand.

### Size growth mechanism from $[Au_{25}(SR)_{18}]^-$ to $[Au_{44}(SR)_{26}]^{2-}$.
To shed light on the size growth from $[Au_{25}(SR)_{18}]^-$ to

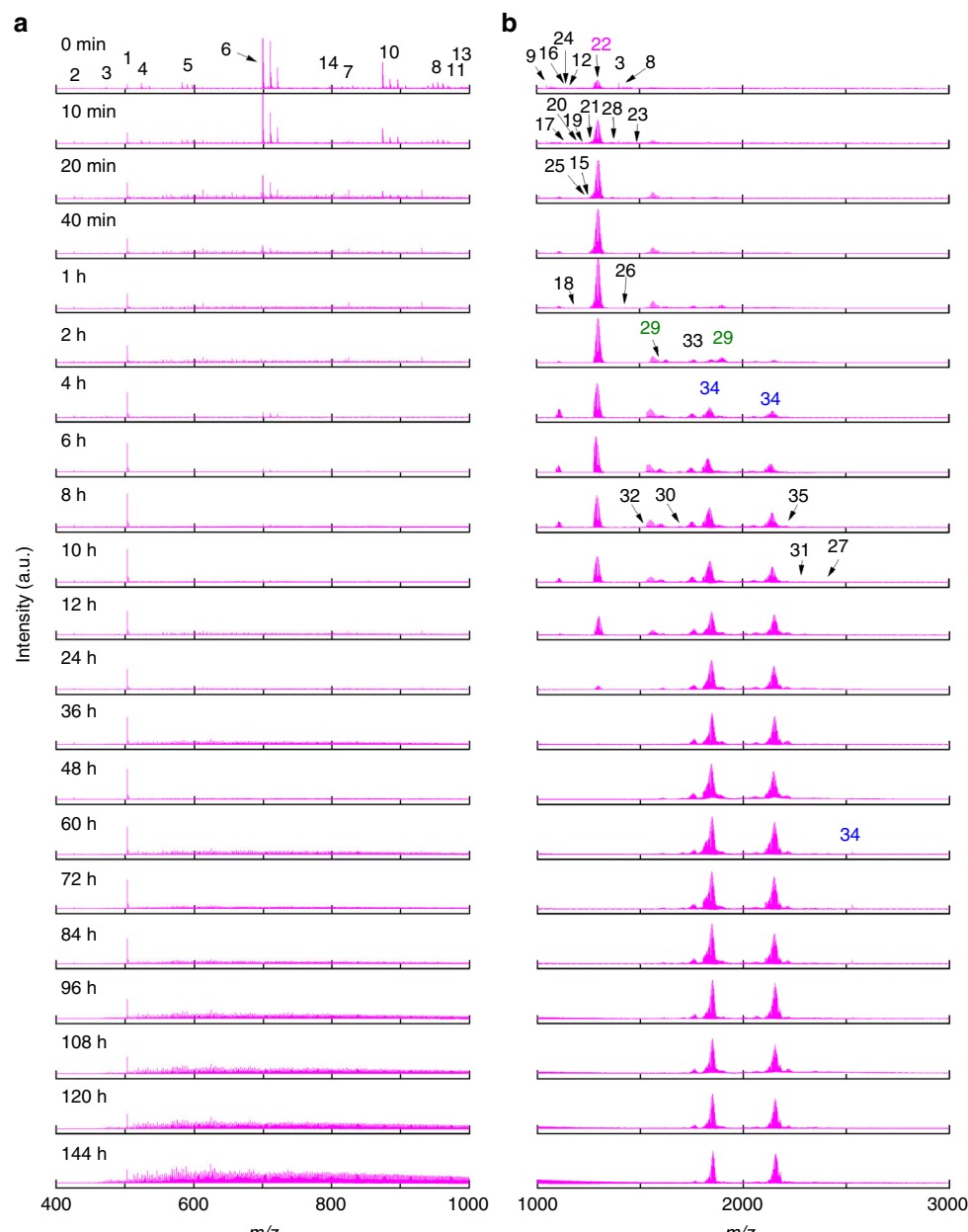

**Fig. 3** Monitoring size growth by electrospray ionization mass spectrometry. Time-course electrospray ionization mass spectra of size growth reaction from $[Au_{25}(SR)_{18}]^-$ to $[Au_{44}(SR)_{26}]^{2-}$ in **a** low and **b** high $m/z$ regime. For ease of identification, the starting cluster $[Au_{25}(SR)_{18}]^-$, product cluster $[Au_{44}(SR)_{26}]^{2-}$, as well as an important intermediate cluster $[Au_{38}(SR)_{24}]^0$ are highlighted in magenta, blue, and olive, respectively. The detailed formula of Au(I)-SR complexes or cluster intermediates identified in mass spectrometry are listed in Table 1

$[Au_{44}(SR)_{26}]^{2-}$, we monitored the reaction process using both UV-vis absorption spectroscopy and ESI-MS. As observed in the time–course UV-vis absorption spectra (Fig. 2), the size growth reaction is apparently divided into three stages. Stage 0 is the "pre–growth stage", which is initiated by bubbling CO into the reaction mixture (right after $t = 0$ min), and this stage lasts through the following 1 h (Fig. 2a). At this stage, the most notable optical feature is an enhancement of the characteristic absorption of $[Au_{25}(SR)_{18}]^-$ (i.e., at 430, 460, and 690 nm), indicating accumulation of $[Au_{25}(SR)_{18}]^-$ by CO reduction of Au(I)-SR complexes. In the following Stage I ($t = 1$–48 h, Fig. 2b), by stark contrast, the absorption features of $[Au_{25}(SR)_{18}]^-$ gradually fade, accompanied by a built-up of absorption features of $[Au_{44}(SR)_{26}]^{2-}$ at 480, 575, and 765 nm. This data suggests that the $[Au_{44}(SR)_{26}]^{2-}$ NCs were formed at the expense of

$[Au_{25}(SR)_{18}]^-$ NCs at this stage, and this stage is referred to as "size growth stage". In the last stage (Stage II, $t = 48$–144 h, Fig. 2c), the absorption features of $[Au_{44}(SR)_{26}]^{2-}$ become better defined, which is in good accordance with a "size-focusing" process[36]. The distinct color change of reddish brown, to dark reddish brown, brown, and finally to dark brown (insets of Fig. 2) also implied a gradual size conversion from $[Au_{25}(SR)_{18}]^-$ to $[Au_{44}(SR)_{26}]^{2-}$.

Although an apparent bottom-up size growth mechanism was indicated by time–course UV-vis absorption spectroscopy, deeper level understandings of the NC growth are limited. To achieve this, we used ESI-MS to identify stable intermediates during the size conversion. Owing to a slow and readily quenchable reduction kinetics made possible by the mild gaseous reductant CO, we were able to identify a total of 35 Au(I)-SR complex/NC

**Table 1 Formula of $[Au_n(SR)_m]^{q}$ [a] species captured in time–course electrospray ionization mass spectra**

| 0 e− | | | | | | 2 e− | | 6 e− | | 8 e− | |
|---|---|---|---|---|---|---|---|---|---|---|---|
| 1 | $[Au(SR)_2]^-$ | 6 | $[Au_6(SR)_6]^0$ | 11 | $[Au_{11}(SR)_{11}]^0$ | 14 | $[Au_{10}(SR)_8]^0$ | 17 | $[Au_{20}(SR)_{14}]^0$ | 19 | $[Au_{21}(SR)_{13}]^0$ |
| 2 | $[Au_2(SR)_3]^-$ | 7 | $[Au_7(SR)_7]^0$ | 12 | $[Au_{13}(SR)_{13}]^0$ | 15 | $[Au_{15}(SR)_{13}]^0$ | 18 | $[Au_{22}(SR)_{17}]^-$ | 20 | $[Au_{24}(SR)_{16}]^0$ |
| 3 | $[Au_4(SR)_4]^0$ | 8 | $[Au_8(SR)_8]^0$ | 13 | $[Au_{14}(SR)_{14}]^0$ | 16 | $[Au_{18}(SR)_{14}]^{2+}$ | | | 21 | $[Au_{25}(SR)_{17}]^0$ |
| 4 | $[Au_4(SR)_5]^-$ | 9 | $[Au_9(SR)_9]^0$ | | | | | | | 22 | $[Au_{25}(SR)_{18}]^-$ |
| 5 | $[Au_5(SR)_5]^0$ | 10 | $[Au_{10}(SR)_{10}]^0$ | | | | | | | 23 | $[Au_{29}(SR)_{21}]^0$ |
| 10 e− | | 12 e− | | 14 e− | | 16 e− | | 18 e− | | 20 e− | |
| 24 | $[Au_{24}(SR)_{14}]^0$ | 26 | $[Au_{33}(SR)_{22}]^-$ | 28 | $[Au_{37}(SR)_{23}]^0$ | 32 | $[Au_{42}(SR)_{25}]^+$ | 33 | $[Au_{43}(SR)_{24}]^+$ | 34 | $[Au_{44}(SR)_{26}]^{2-}$ |
| 25 | $[Au_{28}(SR)_{21}]^{3-}$ | 27 | $[Au_{53}(SR)_{41}]^0$ | 29 | $[Au_{38}(SR)_{24}]^0$ | | | | | 35 | $[Au_{46}(SR)_{27}]^-$ |
| | | | | 30 | $[Au_{40}(SR)_{25}]^+$ | | | | | | |
| | | | | 31 | $[Au_{51}(SR)_{38}]^-$ | | | | | | |

[a] All clusters are captured as anions by the form of $[Au_n(SR)_{m+x}Na_{-y}H]^{q'}$ ($q' < 0$) in ESI-MS analysis (negative ion mode). The net charge of clusters, i.e., $q$ in $[Au_n(SR)_m]^q$ is deduced via the equation $q = q'-(x-y)$.

species at isotope resolution in the entire course of size growth reaction from $[Au_{25}(SR)_{18}]^-$ to $[Au_{44}(SR)_{26}]^{2-}$ (Figs. 3a, b and Table 1). Zoom-in ESI-MS spectra and the isotope patterns of all these species are included in Figs. 1d, e, and Supplementary Figs. 1–33 in Supplementary Information (SI). These Au(I)-SR complex/NC species can be represented by the universal formula $[Au_n(SR)_m]^q$, where their valence electron count can be calculated via the equation: $N^* = n-m-q$. As can be seen in Table 1, these $[Au_n(SR)_m]^q$ species can be roughly divided into two categories: $N^* = 0$ for Au(I)-SR complex species without a Au(0) core, while $N^* > 0$ for Au NC species with a Au(0) core. More importantly, each identified species in the ESI-MS spectra carries an even number of valence electrons ($N^* = 0, 2, 6, 8, 10, 12, 14, 16, 18,$ and 20), which suggests that the clusters grow by 2 e− hopping in size. This finding is similar to our previous observation in the formation of $[Au_{25}(SR)_{18}]^-$, where a 2 e− hopping mechanism was initially attributed to carboxylation-decarboxylation of $[Au_n(SR)_m]^q$-CO adducts (denoted as $[Au_n(SR)_mCO]^q$) featured in the CO-mediated reduction[30]. It should be pointed out that such 2 e− hopping mechanism has also been observed in other reduction systems, indicating that the root cause of the 2 e− hopping mechanism is the relatively good stability of NC species with even-numbered valence electron count[40, 45, 46]. However, due to the lack of X-ray structures of most of the intermediate $[Au_n(SR)_m]^q$ species, the origin of 2 e− hopping mechanism is still under debate and requires further experimental clarification. We also tracked the time-dependent abundance of all 35 $[Au_n(SR)_m]^q$ species (Fig. 4), and presented below are details of such time-dependent ESI-MS analysis, which clearly suggest a three-stage size hopping mechanism similar to the analysis of UV-vis absorption data.

We first examined the reaction mixture prior to CO reduction ($t = 0$ min) by ESI-MS. As shown in Fig. 3 (0 min), the predominant species in the reaction mixture are seed $[Au_{25}(SR)_{18}]^-$ NCs ($N^* = 8$) and Au(I)-SR complex species with $N^* = 0$ (i.e., species 1–13 formulated as $[Au_n(SR)_n]^0$ or $[Au_n(SR)_{n+1}]^-$ with $n \le 14$). Of note, the reaction mixture seems

to be a physical mixture of the as-prepared Au(I)-SR complexes and seed $[Au_{25}(SR)_{18}]^-$ NCs (Supplementary Fig. 34 and Supplementary Table 1). This data also suggests that the size growth reaction is necessarily fueled by CO reduction. Before considering the details of Au(I)-SR complex species in the reaction mixture, it should be noted that the chemical identity and solubility of Au(I)-SR complexes could vary with different intrinsic (e.g., R group) and surrounding (e.g., pH, ionic strength, and solvent polarity) conditions. Based on their various identities and structures, Au(I)-SR complexes species would show varied reactivity towards CO reduction, affecting the cluster growth mechanism. As shown in Fig. 4, $[Au_n(SR)_n]^0$ with $n = 5-14$ (most probably adopt ring-like structures[30, 47]) depleted fast in the 1st h, implying their susceptibility towards CO reduction. In stark contrast, $[Au_n(SR)_{n+1}]^-$ ($n = 1, 2,$ and 4) with linear structures (two ends capped by thiolate ligands) were less reactive or inert towards CO reduction[30], resulting in their persistent survival throughout the entire course of size growth reaction. In addition, a mid-reactive intermediate species towards CO-reduction was identified as $[Au_4(SR)_4]^0$ (lasting longer than those reactive species but shorter than those inert species), which could be due to its exceedingly stable 8-membered-ring structure. It is also worth noting that, despite the aforementioned $[Au_{25}(SR)_{18}]^-$ and Au(I)-SR complex species, minor amounts of other NC species with $N^* > 0$ (e.g., species 14–16 with $N^* = 2$, species 17 with $N^* = 6$, species 19–21 and 23 with $N^* = 8$, and species 24–25 with $N^* = 10$) were also found in the reaction mixture at $t = 0$. The formation of minor amounts of these NC species ($N^* > 0$) could be attributed to the enhanced reduction power of thiolate ligands at elevated pH (13.0), which could reduce Au(III) to Au(0) by boosting the oxidation state of S to +4 or +6[48, 49].

CO was bubbled into the reaction mixture at $t = 0$ min to initiate the size growth. Stage 0 ($t = 0-1$ h) is the pre-growth stage, which features kinetically dictated accumulation of $[Au_{25}(SR)_{18}]^-$ NCs. The most notable changes in this stage are enhancements of the ESI-MS spectral intensities corresponding to $[Au_{25}(SR)_{18}]^-$, accompanied by the diminishing of peaks

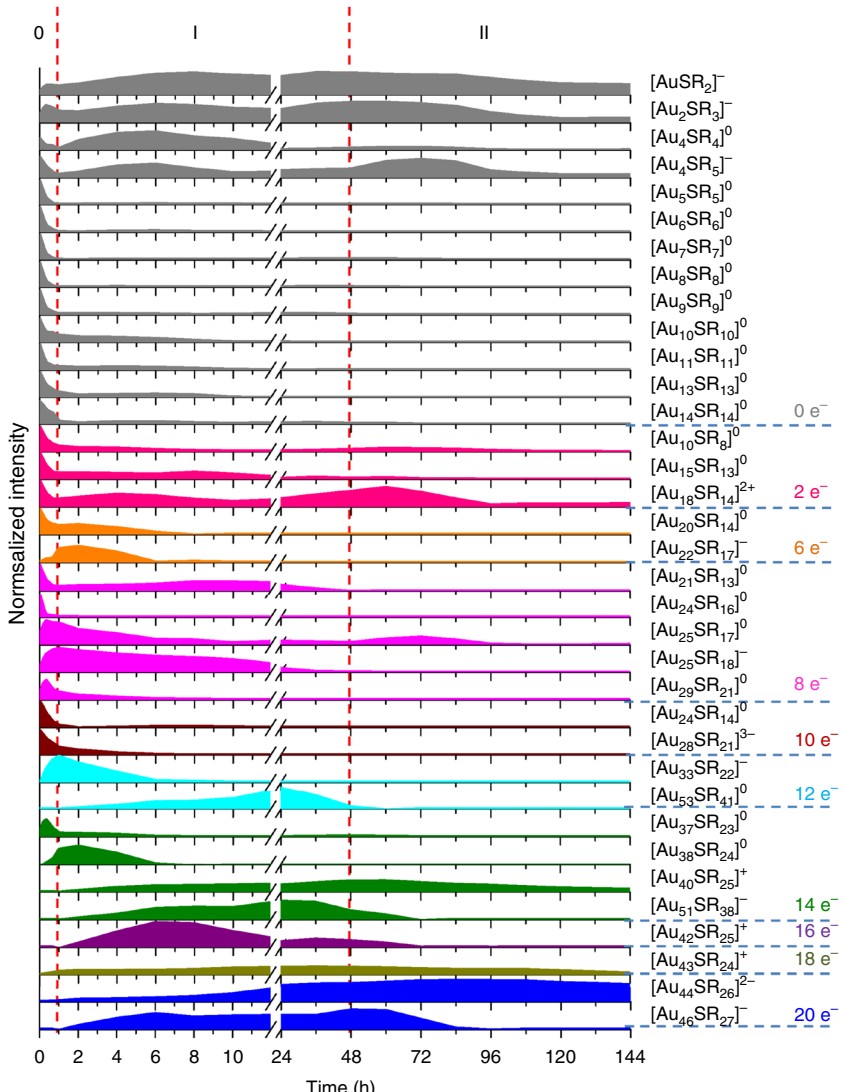

**Fig. 4** Time-dependent abundance of the as-identified Au(I)-SR complex/cluster species. The growth of $[Au_{44}(SR)_{26}]^{2-}$ can be divided into 3 stages: 0) kinetically dictated accumulation of $Au_{25}$, I) $Au_{25}$-mediated size growth, and II) thermodynamically controlled size-focusing

corresponding to reactive Au(I)-SR complex/NC species (Figs. 3 and 4). These observations corroborate well with the findings in UV-vis absorption analysis, and suggest that the conversion of reactive Au(I)-SR complex/NC species into $[Au_{25}(SR)_{18}]^-$ is prompted by CO reduction. Taking both the supreme stability of $[Au_{25}(SR)_{18}]^-$ and susceptibility of reactive Au(I)-SR complex species into account, we proposed that CO would first react with reactive Au(I)-SR complexes through a preferential adsorption. As illustrated in Fig. 5a, this preferred reaction is driven by the lower activation energy ($E_a$) of the reactive Au(I)-SR complex species to form $[Au_n(SR)_nCO]^0$ intermediates over CO complexing with $[Au_{25}(SR)_{18}]^-$ (i.e., formation of $[Au_{25}(SR)_{18}CO]^-$ intermediate). The reduction could then occur by transferring 2 e$^-$ from CO to the complex species via a carboxylation-decarboxylation mechanism[30], generating the NC species with $N^* = 2$. Further reduction of these 2 e$^-$ NC species into 4 e$^-$, 6 e$^-$, 8 e$^-$, and 10 e$^-$ NC species would occur via a similar reduction-growth mechanism. It should be pointed out that we did not capture the 4 e$^-$ NC species in our ESI-MS analysis, most probably due to their short lifetime. Since the supreme stability of $[Au_{25}(SR)_{18}]^-$ suggests this NC species as a distinctive local minimum in the energy landscape of $[Au_n(SR)_m]^q$, the as-formed mixture of $[Au_n(SR)_m]^q$ with $N^*$ ranging from 2 to 10 would

ultimately evolve into $[Au_{25}(SR)_{18}]^-$ in this stage. In addition to the aforementioned reduction-growth, reactions involved in the size evolution into $[Au_{25}(SR)_{18}]^-$ may also include isoelectronic addition, comproportionation, and isoelectronic etching, which are in good agreement with our previous findings[30]. More details about the proposed size evolution mechanism are included in Supplementary Fig. 35 and Supplementary Notes 1 and 2. In contrast to the formation of $[Au_{25}(SR)_{18}]^-$, the consumption pathway of $[Au_{25}(SR)_{18}]^-$ was largely inhibited in this stage due to the relatively higher formation energy obstacle of $[Au_{25}(SR)_{18}CO]^-$ (Fig. 5a). The dynamics of formation and consumption reactions of $[Au_{25}(SR)_{18}]^-$ thereby gave rise to a kinetic accumulation of $[Au_{25}(SR)_{18}]^-$ in this stage.

The successive 1–48 h is Stage I, where a step-wise growth of $[Au_{25}(SR)_{18}]^-$ (8 e$^-$) into 10 e$^-$ (e.g., $[Au_{28}(SR)_{21}]^{3-}$), 12 e$^-$ (e.g., $[Au_{33}(SR)_{22}]^-$ and $[Au_{53}(SR)_{41}]^0$), 14 e$^-$ (e.g., $[Au_{37}(SR)_{23}]^0$, $[Au_{38}(SR)_{24}]^0$, $[Au_{40}(SR)_{25}]^+$, and $[Au_{51}(SR)_{38}]^-$), 16 e$^-$ (e.g., $[Au_{42}(SR)_{25}]^+$), 18 e$^-$ (e.g., $[Au_{43}(SR)_{24}]^+$), and 20 e$^-$ (e.g., $[Au_{44}(SR)_{26}]^{2-}$ and $[Au_{46}(SR)_{27}]^-$) NCs occurs. With the depletion of reactive Au(I)-SR complex/NC species and simultaneous accumulation of $[Au_{25}(SR)_{18}]^-$ in Stage 0, the remaining CO would extensively associate with $[Au_{25}(SR)_{18}]^-$ to form $[Au_{25}(SR)_{18}CO]^-$ intermediate at the beginning of this stage. This

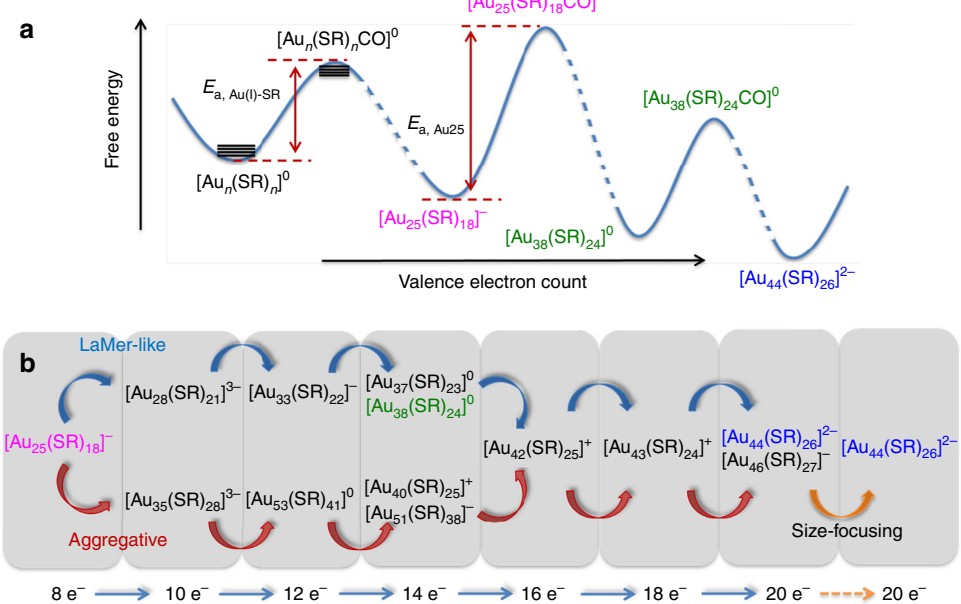

**Fig. 5** Schematic illustration of size growth reaction from $[Au_{25}(SR)_{18}]^-$ to $[Au_{44}(SR)_{26}]^{2-}$. **a** Schematic energy diagram of $[Au_n(SR)_m]^q$ and CO-associated $[Au_n(SR)_m]^q$ species. $E_{a,Au(I)-SR}$ and $E_{a,Au25}$ denotes the formation energy of $[Au_n(SR)_nCO]^0$ and $[Au_{25}(SR)_{18}CO]^-$, respectively; the relative energies are schematically drawn according to the stability of $[Au_n(SR)_m]^q$ species in CO-saturated solution; the dotted curves indicate non-elementary reaction pathways. **b** Growth pathways from $[Au_{25}(SR)_{18}]^-$ to $[Au_{44}(SR)_{26}]^{2-}$

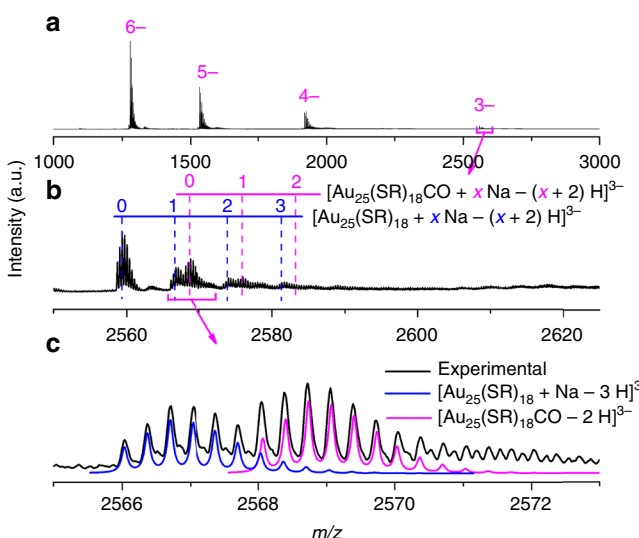

**Fig. 6** Capture of $[Au_{25}(SR)_{18}CO]^-$ intermediate. Electrospray ionization mass spectra of the CO-saturated $[Au_{25}(SR)_{18}]^-$ aqueous solution in **a** large $m/z$ range of 1000–3000, **b** zoom-in $m/z$ range of 2550–2625, and **c** isotope resolution. The dotted drop-lines in **b** are visual guides for $[Au_{25}(SR)_{18}]^-$ (blue lines) and $[Au_{25}(SR)_{18}CO]^-$ (magenta lines) associated with varied numbers of Na⁺. The blue and magenta lines in **c** are simulated isotope patterns of $[Au_{25}(SR)_{18}+Na-3\,H]^{3-}$ and $[Au_{25}(SR)_{18}CO-2\,H]^{3-}$, respectively. The CO-saturated $[Au_{25}(SR)_{18}]^-$ solution is prepared by bubbling CO into an aqueous solution of purified $[Au_{25}(SR)_{18}]^-$ (0.2 mL, [Au] = 10 mM) for 2 min

assertion is experimentally supported by our successful capture of $[Au_{25}(SR)_{18}CO]^-$ intermediate by ESI-MS. As shown in Figs. 6a, b, the zoom-in ESI-MS spectrum of the CO-saturated pure $[Au_{25}(SR)_{18}]^-$ aqueous solution shows representative peaks of $[Au_{25}(SR)_{18}CO]^-$. The isotope analysis depicted in Fig. 6c

further rules out any possible spectral interference from Na⁺-associated $[Au_{25}(SR)_{18}]^-$ NCs, unambiguously corroborating the formation of $[Au_{25}(SR)_{18}CO]^-$. The X-ray crystallography analysis has shown that $[Au_{25}(SR)_{18}]^-$ possesses an icosahedral $Au_{13}$ core[10], twelve of whose triangular $Au_3$ facets are capped by the dimeric SR-Au-SR-Au-SR staple-like motifs, leaving eight $Au_3$ facets uncapped. The uncapped $Au_3$ facet from the icosahedral core together with three Au atoms nearby from the dimeric SR-Au-SR-Au-SR motifs could then form a pocket-like cavity, providing reactive sites for reactant adsorption[21]. Therefore, the formation of $[Au_{25}(SR)_{18}CO]^-$ could be attributed to the accommodation of CO in those pocket-like sites, and in turn the strong binding between CO and Au might make CO more susceptible to oxidation. Further growth of NCs hence depends on the reduction of remaining inert Au(I)-SR complex/NC species by the as-formed reactive $[Au_{25}(SR)_{18}CO]^-$. The crucial role of reactive $[Au_{25}(SR)_{18}CO]^-$ in the proposed size growth mechanism suggests this stage as "$Au_{25}$-mediated size-growth".

The most intriguing finding in the $Au_{25}$-mediated size–growth stage is a dual-mode size growth pattern (Fig. 5b), which is inherently dictated by the co-existence of less reactive Au(I)-SR complex (e.g., $[Au_4(SR)_5]^-$) and NC (e.g., $[Au_{18}(SR)_{14}]^{2+}$) species. The top pathway in Fig. 5b shows a monotonic size growth pattern. This pathway relies on the addition of newly reduced Au(I)-SR complexes on the growing NCs, and thus it could be considered as an analogue of LaMer mechanism. As discussed in Supplementary Note 3, the $[Au_{25}(SR)_{18}CO]^-$ could react with $[Au_4(SR)_5]^-$, leading to the formation of $[Au_{29}(SR)_{23}]^{4-}$ ($N^* = 10$) via the carboxylation-decarboxylation process. The high charge density of $[Au_{29}(SR)_{23}]^{4-}$ would initiate a splitting reaction to generate a more stable $[Au_{28}(SR)_{21}]^{3-}$ species ($N^* = 10$) with a compromised charge density and an ubiquitous $[Au(SR)_2]^-$ ($N^* = 0$). This regeneration mechanism of $[Au_{28}(SR)_{21}]^{3-}$ was supported by the decreased consumption rate of $[Au_{28}(SR)_{21}]^{3-}$ at the beginning of Stage I. The as-formed 10 e⁻ $[Au_{28}(SR)_{21}]^{3-}$ would subsequently evolve into 12 e⁻ (i.e., $[Au_{33}(SR)_{22}]^-$) → 14 e⁻ (i.e., $[Au_{37}(SR)_{23}]^0$ and $[Au_{38}(SR)_{24}]^0$) → 16 e⁻ (i.e., $[Au_{42}(SR)_{25}]^+$) → 18 e⁻ (i.e.,

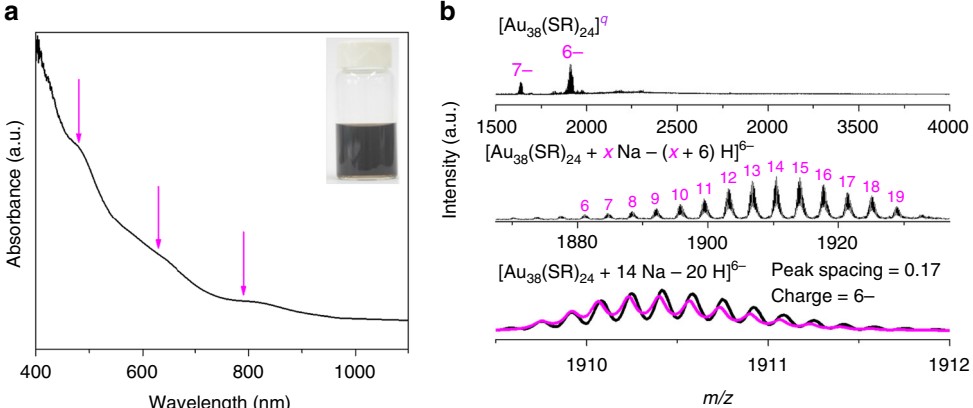

**Fig. 7** Seed-mediated synthesis of $[Au_{38}(SR)_{24}]^0$. **a** Ultraviolet-visible absorption and **b** electrospray ionization mass spectra of $[Au_{38}(SR)_{24}]^0$. The inset in **a** is digital photo of an aqueous solution of $[Au_{38}(SR)_{24}]^0$. The absorption features at 480, 630, and 790 nm in **a** are in good agreement with the most prominent absorption of organic-soluble $[Au_{38}(SR)_{24}]^0$ with a fused bi-icosahedral $Au_{23}$ core[52]. The magenta and black lines in bottom spectra of **b** show the simulated and experimental isotope patterns of $[Au_{38}(SR)_{24}+14\ Na-20\ H]^{6-}$, respectively

$[Au_{43}(SR)_{24}]^+) \rightarrow 20\ e^-$ (i.e., $[Au_{44}(SR)_{26}]^{2-}$ and $[Au_{46}(SR)_{27}]^-$) NCs, via the common reduction-growth mechanism. Such a bottom-up step-wise (with a pace of $2\ e^-$) growth process from 8 to $20\ e^-$ NC species is supported by the temporal appearance sequence of the corresponding NC species (Fig. 4). Detailed formation and consumption pathways of each identified NC species in this pathway are included in Supplementary Discussion.

In addition to bottom-up LaMer-like growth pathway, Fig. 4 also suggests an alternative volcano-shaped growth pattern, where $[Au_{25}(SR)_{18}]^-$ first over-grows into $[Au_{53}(SR)_{41}]^0$ followed by reduction-assisted size-down to $[Au_{44}(SR)_{26}]^{2-}$. As shown in Fig. 5b (bottom pathway), the size evolution follows a detailed pattern of $8\ e^-$ (i.e., $[Au_{25}(SR)_{18}]^-) \rightarrow 10\ e^-$ (i.e., $[Au_{35}(SR)_{28}]^{3-}) \rightarrow 12\ e^-$ (i.e., $[Au_{53}(SR)_{41}]^0) \rightarrow 14\ e^-$ (i.e., $[Au_{40}(SR)_{25}]^+$ and $[Au_{51}(SR)_{38}]^-) \rightarrow 16\ e^-$ (i.e., $[Au_{42}(SR)_{25}]^+) \rightarrow 18\ e^-$ (i.e., $[Au_{43}(SR)_{24}]^+) \rightarrow 20\ e^-$ (i.e., $[Au_{44}(SR)_{26}]^{2-}$ and $[Au_{46}(SR)_{27}]^-$). Specifically, as depicted in Supplementary Note 4, the $[Au_{25}(SR)_{18}CO]^-$ would react with a long Au(I)-SR complex (e.g., $[Au_{10}(SR)_{10}]^0$), giving rise to $[Au_{35}(SR)_{28}CO]^-$ ($N^* = 8$) and eventually evolving into $[Au_{35}(SR)_{28}]^{3-}$ ($N^* = 10$) via the aforementioned carboxylation-decarboxylation mechanism. An aggregation reaction could then occur between the freshly formed $[Au_{35}(SR)_{28}]^{3-}$ and a $[Au_{18}(SR)_{14}]^{2+}$ ($N^* = 2$), yielding a $[Au_{53}(SR)_{41}]^0$ ($N^* = 12$) by ejecting a free $SR^-$. This process resembles the aggregative growth mechanism of NP synthesis, where the agglomeration of primary NPs contributes to the growth of particle size. It is also worth noting that the involvement of long Au(I)-SR complexes in the formation of $[Au_{53}(SR)_{41}]^0$ is indirectly supported by its extraordinarily high SR-to-Au ratio, $R_{SR/Au} = 0.77$, which is distinctly higher than that of the other 12 e- NC species (e.g., $R_{SR/Au} = 0.67$ in $[Au_{33}(SR)_{22}]^-$) and close to that of a family of Au(0)@Au(I)-SR NCs ($R_{SR/Au} = 0.84$) recently reported[18]. Of note, Au(0)@Au(I)-SR NCs consist of a small Au(0) core, with long Au(I)-SR motifs wrapping on its surface.

The size-down of $[Au_{53}(SR)_{41}]^0$ would then occur via two parallel pathways. In the first pathway (Pathway 1, Supplementary Note 5), a $[Au_{53}(SR)_{41}]^0$ ($N^* = 12$) reacts with a CO to produce a $[Au_{51}(SR)_{38}]^-$ ($N^* = 14$) and a ubiquitous $[Au_2(SR)_3]^-$ ($N^* = 0$). Via a similar reduction-assisted size-down reaction, $[Au_{51}(SR)_{38}]^-$ ($N^* = 14$) is then able to downgrade to $[Au_{42}(SR)_{25}]^+$ ($N^* = 16$) by releasing a $[Au_9(SR)_9]^0$ ($N^* = 0$). We rationalize that the reduction-assisted size-down reaction (from $[Au_{53}(SR)_{41}]^0$ to $[Au_{42}(SR)_{25}]^+$) is thermodynamically driven by the fitness of protecting Au(I)-SR motifs to the

curvature of Au(0) core of the NCs[50, 51]. Recent advances in cluster chemistry suggest that long Au(I)-SR motifs could provide better protection to small Au(0) core, as the flexibility of long Au(I)-SR motifs allows them to wrap over the large curvature surface of small Au(0) core without developing significant tension or stress (or vice versa). With the growth of Au(0) core pumped by CO reduction, short Au(I)-SR motifs are preferred by the decreased curvature of Au(0) core, driving the release of long Au(I)-SR motifs (e.g., $[Au_9(SR)_9]^0$) and formation of more compact Au NCs (e.g., $[Au_{42}(SR)_{25}]^+$). Further size evolution of $[Au_{42}(SR)_{25}]^+$ ($N^* = 16$) into $[Au_{43}(SR)_{24}]^+$ ($N^* = 18$), $[Au_{44}(SR)_{26}]^{2-}$ and $[Au_{46}(SR)_{27}]^-$ ($N^* = 20$) species could be readily accomplished via the typical reduction-growth mechanism. In the other parallel pathway (Pathway 2, Supplementary Note 5), the 14 e$^-$ NC species formed by size-down of $[Au_{53}(SR)_{41}]^0$ ($N^* = 12$) is $[Au_{40}(SR)_{25}]^+$ in lieu of $[Au_{51}(SR)_{38}]^-$. This is accomplished via a reduction-splitting process. As illustrated in Supplementary Note 5 (Pathway 2), in the presence of ubiquitous Au(I)-SR complexes such as $[Au(SR)_2]^-$ and $[Au_4(SR)_5]^-$, $[Au_{53}(SR)_{41}]^0$ would be subsequently reduced by two CO yielding $[Au_{58}(SR)_{42}]^0$ ($N^* = 16$), which is unstable due to its high $R_{SR/Au}$ (0.72) and could rapidly split into $[Au_{40}(SR)_{25}]^+$ ($N^* = 14$) and $[Au_{18}(SR)_{14}]^{2+}$ ($N^* = 2$). This mechanism is in good accordance with the accumulation of $[Au_{40}(SR)_{25}]^+$ and $[Au_{18}(SR)_{14}]^{2+}$ at the expense of $[Au_{53}(SR)_{41}]^0$ at the end of Stage I. The growth process of $[Au_{40}(SR)_{25}]^+$ into NCs with higher $N^*$ ($16 \rightarrow 18 \rightarrow 20$) has been discussed previously in Pathway 1, Supplementary Note 5. The volcano-shaped size transition could be considered as an analogue of well-known digestive ripening phenomenon in NP synthesis, where large-sized NPs are converted into small-sized ones via a largely unrevealed mechanism. Therefore, our data suggest that boosting of valence electrons of Au NCs (the value of $N^*$) or the number of Au(0) atoms is a driving force for the digestive ripening phenomenon occurring under the reductive environment in the reaction solution.

At the end of Stage I ($t = 48$ h), $[Au_{44}(SR)_{26}]^{2-}$ has been developed into a predominant species (Figs. 3 and 4), while net formation of other-sized Au NCs has almost ceased (supported by their population concentrations peaking at or before $t = 48$ h). The thermodynamically controlled size-focusing stage (Stage II) then follows the $Au_{25}$-mediated size-growth stage (Stage I). In this stage, other sized Au NCs (e.g., $[Au_{40}(SR)_{25}]^+$, $[Au_{43}(SR)_{24}]^+$, and $[Au_{46}(SR)_{27}]^-$) would be converted into $[Au_{44}(SR)_{26}]^{2-}$ via a slow (~4 days) size-focusing process. The survival of

$[Au_{44}(SR)_{26}]^{2-}$ is due to its supreme stability over NCs of other sizes, while the slowness of the size-focusing reaction is due to high stability of the residual Au NCs. The dominant reactions involved in this stage are therefore reduction-growth and isoelectronic etching, as exemplified in Supplementary Note 6.

**Seed-mediated synthesis of $[Au_{38}(SR)_{24}]^0$.** Based on the above understandings on the size growth mechanism, we have a good opportunity to trap the growth of Au NCs into other stable intermediate sizes by a deliberate control over growth kinetics of Au NCs. One target Au NC is $[Au_{38}(SR)_{24}]^0$, whose stability has been well demonstrated in organic solution[52–55]. Developments in synthetic chemistry for the past two decades have documented numerous synthetic methods for organic-soluble $[Au_{38}(SR)_{24}]^0$ NCs[52–54], but the same could not be said for water-soluble $[Au_{38}(SR)_{24}]^0$. By noting $[Au_{38}(SR)_{24}]^0$ ($N^* = 14$) as an important intermediate species in LaMer-like growth of $[Au_{44}(SR)_{26}]^{2-}$ (Fig. 5b), we hypothesized that water-soluble $[Au_{38}(SR)_{24}]^0$ could be produced by concurrently suppressing the aggregative growth pathway of $[Au_{44}(SR)_{26}]^{2-}$ (bottom pathway, Fig. 5b) and bringing down the reducing power in LaMer-like growth pathway. As the volcano-shaped growth in the aggregative growth pathway requires long Au(I)-SR complexes, we thus proposed that a downward shift of Au(I)-SR complex size, which could be made possible via increasing the feeding ratio of SR-to-Au[18], would help inhibit the aggregative growth pathway. Regarding the reducing power, we have demonstrated previously that lowering the pH of reaction solution is an effective way to decrease the reducing power of CO. By a combined effect of the elevated SR-to-Au ratio (2:1 vs. 1:1 for $[Au_{44}(SR)_{26}]^{2-}$) and decreased pH of the reaction solution (12.4 vs. 13.0 for $[Au_{44}(SR)_{26}]^{2-}$), we have successfully produced atomically precise $[Au_{38}(SR)_{24}]^0$ of high purity (Figs. 7a, b). It is worth pointing out that solely elevating SR-to-Au ratio or tuning down the solution pH could shift the population of product Au NCs downward (Supplementary Figs. 36 and 37), but it was not capable of producing pure $[Au_{38}(SR)_{24}]^0$, highlighting the importance of kinetic control in both LaMer-like growth and aggregative growth. This result is exciting since it evokes the possibility of producing other intermediate NCs by a kinetically trapping method.

**Icosahedron-based NC series.** During the preparation of this manuscript, we noted that the X-ray structure of $[Au_{44}(SR)_{26}]^0$ (SR = 2,4-dimethylbenzenethiol) has been successfully resolved by Wu and co-workers[15]. The $[Au_{44}(SR)_{26}]^0$ possesses a $Au_{29}$ core, which can be regarded as a fused bi-icosahedral $Au_{23}$ core (similar to the core of $[Au_{38}(SR)_{24}]^0$) capped at the bottom with a boat-like $Au_6$ module. The $Au_{29}$ core is protected by three monomeric SR-Au-SR motifs at the bottom and six dimeric SR-Au-SR-Au-SR motifs at the top and waist. As the UV-vis absorption spectra of $[Au_{44}(SR)_{26}]^0$ and $[Au_{44}(SR)_{26}]^{2-}$ share a similar envelope with several minor differences (less distinct and red-shifted absorption features in the latter), $[Au_{44}(SR)_{26}]^{2-}$ should share a similar $Au_{44}S_{26}$ (most probably with a slight twist in $Au_{29}$ core caused by 2 e$^-$ core charge) frame with $[Au_{44}(SR)_{26}]^0$. To further confirm this point, we simulated the optical absorption spectrum of $[Au_{44}(SR)_{26}]^{2-}$ based on the experimental structure of $[Au_{44}(SR)_{26}]^0$ but for a dianion charge state with density functional theory (DFT) structure optimization. The time-dependent DFT (TDDFT) simulation at the B3LYP level yielded a spectrum (Supplementary Fig. 38) showing several distinct bands in reasonable agreement with the experiment; the differences between the simulation and the experimental results could be due to simplified functional and basis set, and different ligands (-SCH$_3$ used for –SR) used in our simulation. We also

computed the TDDFT spectrum of the neutral $[Au_{44}(SR)_{26}]^0$ (Supplementary Fig. 38), whose worse agreement with experiment supported the 2- charge state in as-obtained Au$_{44}$ NCs.

The available structure model for $[Au_{44}(SR)_{26}]^{2-}$ (20 e$^-$) now allows us to compute its relative stability to $[Au_{25}(SR)_{18}]^-$ (8 e$^-$). By constructing a Hess reaction of two $[Au_{25}(SR)_{18}]^-$ clusters (16 e$^-$) reduced by two CO molecules (4 e$^-$) to form a $[Au_{44}(SR)_{26}]^{2-}$ (20 e$^-$) in an aqueous solution, we found that the reaction has an energetic change of $-200$ kcal per mol of $[Au_{25}(SR)_{18}]^-$, indicating that $[Au_{44}(SR)_{26}]^{2-}$ is indeed more stable than $[Au_{25}(SR)_{18}]^-$ in the presence of CO in solution (Supplementary Note 7). The high stability of $[Au_{44}(SR)_{26}]^{2-}$ is attributed to its highly symmetric atomic packing pattern as well as 20 e$^-$ shell-closing electronic configuration ($1S^2 1P^6 1D^{10} 2S^2$).

The three most stable sizes identified in this study, i.e., $[Au_{25}(SR)_{18}]^-$ (icosahedral $Au_{13}$ core)[10, 56], $[Au_{38}(SR)_{24}]^0$ (fused bi-icosahedral $Au_{23}$ core)[55], and $[Au_{44}(SR)_{26}]^{2-}$ (bottom-capped bi-icosahedral $Au_{29}$ core)[15], constitute a new magic size series based on the icosahedral $Au_{13}$ unit, which are distinctly different from the face-centered cubic (FCC)-based magic size series, $Au_{8N+4}(SR)_{4N+8}$ (e.g., $Au_{28}(SR)_{20}$, $Au_{36}(SR)_{24}$, $Au_{44}(SR)_{28}$, $Au_{52}(SR)_{32}$, and $Au_{76}(SR)_{44}$) recently revealed by both experimental and theoretical means[57, 58]. The cluster cores in the latter are constructed by successively packing $Au_8$ layer in (001) direction, which is vastly different from fusion of the icosahedral $Au_{13}$ units in the former. The distinctly different structure construction manners indicate that the size of Au NCs might evolve in diverse routes in the sub-2 nm regime, and such diversity in size evolution may provide additional opportunities for engineering cluster properties for practical applications.

## Discussion

In summary, we have developed a seed-mediated growth method for synthesis of high quality $[Au_{44}(SR)_{26}]^{2-}$ from $[Au_{25}(SR)_{18}]^-$. The size growth is dependent on a three-stage size hopping mechanism: Stage 0, kinetically dictated accumulation of Au$_{25}$; Stage I, Au$_{25}$-mediated size growth; and Stage II, thermodynamically controlled size-focusing. The accumulation of $[Au_{25}(SR)_{18}]^-$ in Stage 0 is directed by varied reactivity of Au(I)-SR complexes and $[Au_{25}(SR)_{18}]^-$ towards CO reduction. With a systematic investigation of the formation and consumption reactions of 35 Au(I)-SR complex/NC species captured in the ESI-MS spectra, we identified the initiation (i.e., adsorption of CO on $[Au_{25}(SR)_{18}]^-$), driving force (i.e., 2 e$^-$ boosting valence electron counts), and detailed size evolution patterns (i.e., LaMer-like and aggregative growth) for the size growth reactions from $[Au_{25}(SR)_{18}]^-$ to $[Au_{44}(SR)_{26}]^{2-}$. Based on these molecular-level insights into cluster/particle growth, we were also able to drive the seed-mediated growth reaction kit to produce intermediate sizes (e.g., molecularly pure $[Au_{38}(SR)_{24}]^0$). The seed-mediated growth method and detailed mechanism study presented in this work are of interest not only because it provides an effective way to trap NC growth into a number of desirable sizes, but also because it offers new insights into some fundamentals at molecular level, such as LaMer mechanism, aggregative growth, and digestive ripening, which have puzzled the nanoscience and nanomaterials research communities for decades.

## Methods

**Materials.** Hydrogen tetrachloroaurate (III) trihydrate (HAuCl$_4$·3H$_2$O), *para*-mercaptobenzoic acid (*p*-MBA), and *N,N*-dimethylformamide (DMF) from Sigma Aldrich; sodium hydroxide (NaOH) from Merck; ethanol from Fisher; and carbon monoxide (CO, 99.9%) from Singapore Oxygen Air Liquide Pte Ltd. (SOXAL) were used as-received without further purification. Ultrapure Millipore water (18.2 MΩ•cm) was used in the preparation of all aqueous solutions. All glassware were washed with aqua regia and rinsed with ethanol and ultrapure water before use.

**Synthesis of [Au$_{25}$(*p*-MBA)$_{18}$]$^-$.** [Au$_{25}$(*p*-MBA)$_{18}$]$^-$ NCs were prepared according to a reported protocol with several minor amendments[30]. In particular, 10 mL of aqueous solution of 50 mM *p*-MBA (in 150 mM NaOH) and 5 mL of aqueous solution of 50 mM HAuCl$_4$ were added into 238.75 mL of ultrapure water in sequence, and the reaction mixture was stirred at 1,000 r.p.m. for 5 min. The pH of reaction mixture was then brought up to 10.5 by dropping in 1 M NaOH aqueous solution. After stirring for another 30 min, a light-yellow solution of Au(I)-(*p*-MBA) complexes was formed. Subsequently, CO was bubbled into the reaction mixture for 2 min to initiate the reduction of Au(I)-(*p*-MBA) complexes. The reaction was allowed to proceed air-tightly for 3 days at room temperature (25 °C) and under vigorous stirring (1,000 r.p.m.). The reddish brown solution obtained at the end of this procedure was collected as raw product.

The raw product was first concentrated by 10 times via rotary evaporation (water bath temperature 40 °C, cooling temperature 4 °C, and rotation rate 160 r.p.m.). Ethanol (double the volume of the concentrated NC solution) was then added, followed by a centrifugation at 14,000 r.p.m. for 5 min. The resultant pellet was washed with DMF for 2 times and re-dissolved in ultrapure water to form an aqueous solution of purified [Au$_{25}$(*p*-MBA)$_{18}$]$^-$ ([Au] = 10 mM) for further use.

**Seed-mediated synthesis of [Au$_{44}$(*p*-MBA)$_{26}$]$^{2-}$.** 0.25 mL of 50 mM *p*-MBA ethanolic solution and 0.25 mL of 50 mM HAuCl$_4$ aqueous solution were added into 9 mL of ultrapure water in sequence, followed by stirring at 1,000 r.p.m. for 5 min to form a pale-yellow suspension of Au(I)-(*p*-MBA) complexes. The pH of reaction mixture was then brought up to 13.0 by dropping in 1 M NaOH, which turned the pale-yellow suspension into light-yellow solution. The reaction mixture was stirred for another 30 min prior to the addition of 0.25 mL of purified [Au$_{25}$(*p*-MBA)$_{18}$]$^-$ aqueous solution (see above for the detailed preparation). Subsequently, CO was bubbled into the reaction mixture for 2 min to initiate the growth of NCs. After stirring for 6 days at room temperature (25 °C), a black-brown solution was obtained as raw product. Cleaned [Au$_{44}$(*p*-MBA)$_{26}$]$^{2-}$ was obtained by centrifuging a mixture of raw product and ethanol (1/5 V/V) at 14,000 r.p.m. for 5 min, followed by a DMF washing (2 times) and re-dissolution in ultrapure water.

**Seed-mediated synthesis of [Au$_{38}$(*p*-MBA)$_{24}$]$^0$.** The synthetic protocol of [Au$_{38}$(*p*-MBA)$_{24}$]$^0$ is similar to that of [Au$_{44}$(*p*-MBA)$_{26}$]$^{2-}$, except for two minor changes. The first change is the feeding amount of *p*-MBA solution, which is 0.5 mL in the synthesis of [Au$_{38}$(*p*-MBA)$_{24}$]$^0$ rather than 0.25 mL in the synthesis of [Au$_{44}$(*p*-MBA)$_{26}$]$^{2-}$. The other change is the solution pH, which was maintained at 12.4 in the synthesis of [Au$_{38}$(*p*-MBA)$_{24}$]$^0$, while the solution pH was 13.0 in the synthesis of [Au$_{44}$(*p*-MBA)$_{26}$]$^{2-}$.

**Density functional theory computation.** Parallel, resolution-of-identity density functional theory (DFT) calculations with the TPSS form of the meta generalized gradient approximation (meta-GGA) for electron exchange and correlation[59] and the def2-SV(P) basis sets were performed with the quantum chemistry program Turbomole V6.5[60]. Effective core potentials which have 19 valence electrons and include scalar relativistic corrections were used for Au[61]. The Conductor-like Screening Model (COSMO)[62] implemented in Turbomole was used to compute the energies of solvated species. Time-dependent DFT calculation of the UV-vis absorption spectra was done at the B3LYP level. All transitions together with their oscillator strengths were then convoluted with a Lorentzian line shape of 0.15 eV broadening to make the whole optical absorption spectrum.

**Materials characterizations.** Solution pH of reaction mixture was measured by a Mettler-Toledo FE 20 pH-meter. UV-vis absorption spectra were recorded on a Shimadzu UV-1800 spectrometer. The absorbance of reaction mixture was reset (to zero) and made reference to that of ultrapure water in individual test. ESI-MS spectra were obtained on a Bruker microTOF-Q system in negative ion mode. Detailed operating conditions of ESI-MS analysis are given as following: source temperature 120 °C, dry gas flow rate 8 L per min, nebulizer pressure 3 bar, capillary voltage 3.5 kV, and sample injection rate 3 μL per min. In a typical ESI-MS analysis, 0.2 mL of NC solution ([Au] = ~2.5 mM) was mixed with 1 mL ethanol, followed by centrifugation at 14,000 r.p.m. for 5 min. After discarding the supernatant, the resultant pellet was washed by 1 mL DMF, and then re-dissolved in 0.4 mL ultrapure water for ESI-MS measurement. The time-dependent ESI-MS spectra were normalized to the total ion count in individual test.

**Data availability.** All relevant data are available from the corresponding author on request.

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

## Acknowledgements

We acknowledge the financial support from the Ministry of Education, Singapore, Academic Research Grant R-279-000-481-112. V.F. and D.-e.J. were supported by the Division of Chemical Sciences, Geosciences and Biosciences, Office of Basic Energy Sciences, U.S. Department of Energy.

## Author contributions

Q.Y. and X.Y. contributed equally to this work. This project was supervised by J.X. Q.Y., X.Y. and J.X. designed the experiments. Q.Y., X.Y. and Y.Y. conducted experiments, and V.F. and D.-e.J. performed the DFT calculation. Discussion of results and manuscript preparation are group efforts of Q.Y., X.Y., V.F., Y.Y., D.T.L., D.-e.J. and J.X.

## Additional information

**Competing interests:** The authors declare no competing financial interests.

