## [Peer Review File · Nature Communications]

Reviewers' comments:

Reviewer #1 (Remarks to the Author):

The authors take a 'total synthesis' approach for describing the seeded growth of nanoparticles. The work describes in great detail and atomic precision how the seeded growth of gold nanoclusters evolves from Au₂₅ and Au₄₄. By doing so the authors connect classical growth theories to the molecular steps leading to the formation of the product nanoparticle. A key finding by the authors is that the gold nanocluster system evolves in steps between species that differ by two electrons. I believe this study will have broad implications for understanding the nucleation and growth of nanoparticles. I recommend the publication of this work in Nature Communications after the authors address the following minor points:

- 1) The gold-thiol complex can exist in multiple states. The authors should use an appropriate method such as NMR to identify the states of the complexes in the starting solution.
- 2) The purity of the initial state and final state, as well as some intermediate states of the nanocluster (e.g. Au₃₈) should be checked by gel-electrophoresis or similar methodology.
- 3) Do the authors observe any reverse ripening as was observed in silver nanoclusters
10.1021/acs.chemmater.5b00650
- 4) There is a significant mismatch between the simulated and experimental spectra shown in supplementary Figures 22 to 27.

Reviewer #2 (Remarks to the Author):

Thiolate-protected gold clusters are currently in the focus of research. Such clusters can serve as model systems to clarify fundamental issues concerning the nature of nanoscale objects and their properties. The present contribution aims at getting fundamental insight on the growth mechanism of gold nanoclusters. This topic is a very important one, as the better understanding could lead to a more rational design of synthesis strategies. In this respect the topic is very appealing and of general interest to many researchers. Despite this I have to confess that I did not learn much new from reading the manuscript. For sure there are some nice points but I definitely expected more. What is demonstrated is that one stable cluster is transformed into another one (, which has been shown before) and that many "intermediate" species are observed underway. A lot remains speculative as already indicated by the language used by the authors (e.g. "...which indicates that the size growth most probably occurs via a 2 e-hopping mechanism."). Furthermore, the importance of the work is diminished by the fact that the reduction was performed by CO, which is a quite special case and there remains always the question whether the growth mechanism would be the same for other, more commonly used reducing agents. For these reasons my feeling is that the physical insight that can really be drawn from the work does not meet the high expectations that the reader has concerning manuscripts published in Nature Communications.

Some more specific points:

- The quality of the figures is not sufficient (e.g. in Figure 1 the numbers in the spectra cannot be read), but this maybe a problem of reproduction / resolution.
- In Figure 2 the spectra are definitely too small, in c) the lines are not distinguishable at all.
- The authors start with Au₂₅ and then reduce more Au-SR to form further Au₂₅. Is the initial Au₂₅ present important in this step? If the reactions starts from just the Au-SR complex is Au₂₅ formed as

well?

-Au₄(SR)₄ is often observed as a fragment in mass spectrometry. Can the authors comment on that.
-The authors bubbled CO through the solution to initiate the reaction, then the reaction proceeded in an "air tight" container. This means that initially there was a lot of CO and maybe at a later stage there was only little or none. Can the authors comment on that?

Reviewer #3 (Remarks to the Author):

The authors are studying growth mechanisms of gold-thiolate nanoparticles. They have an approach that uses CO reduction, which occurs slowly enough that they can detect intermediates using ESI-MS. This approach is not used often enough in the community, and definitely has the ability to provide insights into growth mechanisms. There are a number of places where I think the manuscript needs to be improved before being considered ready for publication. The manuscript seems quite long, and I am not sure if Nature Communications is the correct place for it. Nature Chemistry or JACS might be more appropriate.

Introduction

The authors only cite a previous study by their group about developing a microscopic understanding of growth mechanisms, which might suggest to the reader that only one study has been performed in the past. The authors should include some of the seminal work by the Lennox and Aikens groups that have aided our understanding of microscopic growth mechanisms, including studies on two electron reduction for converting Au(I)SR complexes.

Results

Numbers in Figure 1 are not readable even when I zoom in on my computer. Resolution in Figure 2 is even worse. Figure 5 needs improvement as well. Both size and resolution need improvement.

I would think that the authors would see isobestic points in their spectra in Figure 2b (and possibly 2c). Do they need to reconsider spectral normalization? (How are the UV-Vis spectra normalized in this work?)

Rephrase the line "The corresponding species labeled are formulated in Table 1", as it is not clear what is meant by this sentence.

Structure "9" seems to be absent in Figure 3, but is listed in Table 1 (and it is later stated in the text that the 35 structures in Table 1 were observed). On the other hand, [Au₁₂(SR)₁₂]₀ is left out of Table 1 (which may be because it does not appear in Figure 3). It is surprising that all of the Au_n(SR)_n appear in Table 1 except for n=12. Also, many of the numbered species in Table 1 are listed as neutral, but they show up in the ESI-MS spectrum. It is only clear much later in the paper when the authors refer to supplementary figures 1-33 that the formulas observed in the ESI-MS are not exactly those listed in the table (varying by H⁺, Na⁺, etc.). It would be good to mention the supplementary figures earlier (and probably it would be good to make it clear that the charges are not the same as those listed in the table). There is some color coding of structure numbers in Figure 3 and Table 1, but it is not explained in either caption. Structure 22 moves in the fourth line down. I think this might be an error. Anyhow, this figure and table need some work.

It would be good to have ESI-MS of the reaction mixture BEFORE CO addition (when it is just Au(I)-SR complexes prepared as stated on p. 7). The CO reactions may be so fast that it is hard to know what to attribute to the initial reaction mixture and what occurs within a very short time (minutes) after CO

is added. For example, the Au₂₅ system must grow very quickly after CO is added (according to the text) without intermediates with Au_n (n > ~15). It seems likely that some intermediates are too reactive to be observed. However, I don't think the authors should claim that they have seen "all" intermediates since they may have missed some (just because they can not be observed); they point this out on p. 14, but it is at odds with their abstract (which says they have seen all steps). The number of intermediates they have seen is great, though!

Observation of even number of valence electrons could imply a 2 electron mechanism, or that odd electron species are very reactive (and thus not observed). It would be good if the authors mention the last point, even if they lean toward the 2 electron mechanism. Moreover, I would not attribute the observation to a "network of superatoms".

In SI, p. 39, the authors mention a proposed decarboxylation of Au₁₀(SR)₁₀COO, but the product they mention has also lost two SR groups. This is not discussed. Things listed on p. 39 are very hypothetical (but it is nice that the authors are thinking about these possibilities). But, they should state that the reactions involved in the size evolution into Au₂₅(SR)₁₈- "may include" (not "also include") ... and that it is a "proposed" size evolution mechanism (main text, p. 14). Be careful not to overstate your findings if the mechanism is very much a hypothesis.

Figure 5 shows one option of Au₂₅(SR)₁₈- undergoing aggregative growth to Au₃₅(SR)₂₈³⁻. This would mean addition of Au₁₀(SR)₁₀²⁻, which was not mentioned as an intermediate. It is eventually discussed on p. 18, but the charge changes and no hypothesis is given.

The solubility of the Au(I)-SR complexes has been questioned and appears to depend on the R group, solvent, and pH. The authors use high pH and water-soluble R groups, which is known to increase the solubility of the Au(I)-SR complexes. However, the things learned under these conditions may not apply to other conditions (especially ones in which the Au(I)-SR complexes are not as soluble). These considerations should be briefly discussed in the manuscript.

The purpose of Figure 5 is not clear. 5a was mentioned in the text (but not really explained).

p. 16 [Au₁₈(SR)₁₄]²⁺ is mentioned. How could the positively charged species (especially with 2+) form under these basic reaction conditions? The plausibility of this should be discussed.

p. 16 "should be attributed" should be rephrased ("could be attributed?"), since it is referring to something that is still very much a hypothesis.

p. 17 "is proven" should be "is supported". Again, don't overstate your findings.

p. 18 The authors have not explained why the aggregative growth pathway is "volcano-shaped". I infer that it is because they suggest the particles get bigger before they get smaller, but the term volcano is used throughout chemistry in a variety of contexts, and should be explained.

The authors could consider doing a kinetic analysis of their system to see if these mechanisms are supported by the available information. This would be difficult (and certainly not expected for the current paper), but it really could help support or refute some of the hypotheses they have that can not currently be supported in detail.

The authors considered only changing the SR-to-Au ratio or the pH of the solution and said they do not achieve the Au₃₈ species. However, the supplementary figure 35 appears to have Au₃₈ in it (where only the SR-to-Au ratio is changed). Is this correct?

The TDDFT spectrum for the neutral Au₄₄ system should also be presented so that the differences between theory and experiment for this system with this level of theory can be assessed. The spectra presented in Supplementary figure 37 differ a fair bit. The methods section says def2-TZVP, but the SI section says def2-SV(P). Could the authors change one of these?

Overall comments

The authors should be commended for tackling a difficult problem that has the potential to shed light on growth mechanisms. However, there is currently a tendency to overstate their hypotheses as “mechanisms” without direct support.

In a number of places in the manuscript, there is some awkward phrasing. You may wish to go through and wordsmith the document some more to remove these.

Replies to reviewers' comments and descriptions of revisions made

Comments by Reviewer #1:

The authors take a 'total synthesis' approach for describing the seeded growth of nanoparticles. The work describes in great detail and atomic precision how the seeded growth of gold nanoclusters evolves from Au₂₅ and Au₄₄. By doing so the authors connect classical growth theories to the molecular steps leading to the formation of the product nanoparticle. A key finding by the authors is that the gold nanocluster system evolves in steps between species that differ by two electrons. I believe this study will have broad implications for understanding the nucleation and growth of nanoparticles. I recommend the publication of this work in Nature Communications after the authors address the following minor points:

Reply: We are glad that the reviewer finds this work interesting and important. We share the reviewer's view that our findings deepen current understandings on the seed-mediated growth of nanoparticles (NPs) and will stimulate more fundamental and applied research on functional NPs. We also thank the reviewer's detailed technical comments/suggestions, which we have addressed point-by-point below.

Detailed comments:

1. The gold-thiol complex can exist in multiple states. The authors should use an appropriate method such as NMR to identify the states of the complexes in the starting solution.

Reply: We are in complete agreement with the reviewer that gold-thiolate (Au(I)-SR) complexes could exist in multiple states. To identify them, we used electrospray ionization mass spectrometry (ESI-MS) to examine the starting Au(I)-SR complexes (Supplementary Figure 34b). The Au(I)-SR complexes are mostly a mixture of cyclic [Au_n(SR)_n]⁰ or linear [Au_n(SR)_{n+1}]⁻ species with varied $n \leq 14$. As the reviewer suggested, we also tried NMR; but due to the polydispersed nature of Au(I)-SR complexes, the NMR analysis is less informative than ESI-MS, so we did not include that in the present work. However, precisely controlling the size of Au(I)-SR complexes and using them in cluster synthesis is another ongoing project in our lab, where NMR analysis instead will be more informative.

Revisions:

Supplementary Information (SI), Pages 35-36, Supplementary Figure 34 and Supplementary Table 1:

More details about the chemical identities of starting Au(I)-SR complexes from ESI-MS have been included.

2) The purity of the initial state and final state, as well as some intermediate states of the nanocluster (e.g. Au₃₈) should be checked by gel-electrophoresis or similar methodology.

Reply: As the reviewer suggested, we have checked the purity of starting cluster (i.e., $[\text{Au}_{25}(\text{SR})_{18}]^-$), final product (i.e., $[\text{Au}_{44}(\text{SR})_{26}]^{2-}$) and featured intermediate (i.e., $[\text{Au}_{38}(\text{SR})_{24}]^0$) by polyacrylamide gel electrophoresis (PAGE). As can be seen in Figure CL-1, only one band could be found in PAGE gel of each cluster, indicating the high purity of as-produced clusters. The PAGE results are also consistent with the distinct peaks in the mass spectra of corresponding clusters (Figures 1d, 1e and 7b).

Figure CL-1. PAGE results of $[\text{Au}_{25}(\text{SR})_{18}]^-$ (left), $[\text{Au}_{38}(\text{SR})_{24}]^0$ (middle) and $[\text{Au}_{44}(\text{SR})_{26}]^{2-}$ (right) NCs. The resolving gel was prepared by 30 w.t.% of acrylamide monomer with a running voltage and time of 160 V and 2.5 h, respectively.

3) Do the authors observe any reverse ripening as was observed in silver nanoclusters 10.1021/acs.chemmater.5b00650.

Reply: We did not intentionally look out for reverse ripening in this work, and our main findings suggested its negligible role in the present work. However, the reviewer's insightful comment inspired us to carry out a reverse ripening reaction upon $[\text{Au}_{38}(\text{SR})_{24}]^0$ NCs *under oxidative environment*. O_2 was bubbled into a solution of freshly prepared $[\text{Au}_{38}(\text{SR})_{24}]^0$ NCs for 10 min at pH 10.5, facilitating etching reaction (in a time course of 6 days) of $[\text{Au}_{38}(\text{SR})_{24}]^0$ by the residual Au(I)-SR complexes. As shown in Figure CL-2, the resultant NCs possess a broad size ($n = 25-44$) and valence electron count ($N^* = 8-16$, where $N^* = n - m - q$ was calculated based on the formula $[\text{Au}_n(\text{SR})_m]^q$) distribution. The formation of smaller-sized $[\text{Au}_{25}(\text{SR})_{18}]^-$ (dominant species) and $[\text{Au}_{36}(\text{SR})_{24}]^0$ is similar to the reported reverse ripening of Ag NCs, where $\text{Ag}_{44}(\text{SR})_{30}$ was converted into $\text{Ag}_{35}(\text{SR})_{18}$ via some intermediate sizes. Intriguingly, in addition to the smaller-sized $[\text{Au}_{25}(\text{SR})_{18}]^-$ and $[\text{Au}_{36}(\text{SR})_{24}]^0$, some larger-sized species like $[\text{Au}_{41}(\text{SR})_{28}]^-$ and $[\text{Au}_{44}(\text{SR})_{28}]^0$ could also be observed in the product. The formation process of those larger-sized NCs most probably involves disproportionation reaction (*J. Am. Chem. Soc.* **2014**, *136*, 10577) of $[\text{Au}_{38}(\text{SR})_{24}]^0$. However, we would like to mention that the data present in Figure CL-2 is quite preliminary, and more should be done to reveal a detailed reaction mechanism between atomically precise clusters and Au(I)-SR complexes under the stated mildly oxidative condition. Due to the as-demonstrated similar behaviors between Au and Ag NCs, the reader of this paper may also be interested in the size/composition manipulation of Ag NCs (*Chem. Mater.* **2015**, *27*, 4289; *J. Am. Chem. Soc.* **2014**, *136*, 15865), which have been cited.

Figure CL-2. Electrospray ionization mass spectrum of $[\text{Au}_{38}(\text{SR})_{24}]^0$ reacted with Au(I)-SR complexes in O_2 -saturated water at a solution pH of 10.5.

4) There is a significant mismatch between the simulated and experimental spectra shown in supplementary Figures 22 to 27.

Reply: We thank the reviewer for his/her careful critique, which has spurred us to re-examine our assignment of the ESI-MS spectra. We agree that relatively large discrepancy between the simulated and experimental isotope patterns was observed in as-mentioned mass spectra. This should be attributed to the relatively low signal-to-noise ratio of these intermediate species. We would also like to point out that the mismatch between the simulated and experimental formula of all identified Au(I)-SR complex or NC species is within 0.5 Da, supporting the good accuracy of our assignment. Such mismatch (0.5 Da or even higher) is commonly found in the reported mass spectra of thiolate-protected metal NCs (e.g., *Nat. Commun.* **2015**, 6, 8667; *J. Am. Chem. Soc.* **2014**, 137, 1206; *J. Phys. Chem. Lett.* **2014**, 5, 3757). On the other hand, we still took heed of the critique and persist (in this revision) to produce better-quality isotope patterns, by substituting low signal-to-noise peaks with better defined ones (please see revised Supplementary Figures 25-27). We believe that this revision has narrowed the gap between simulated and experimental information.

Revisions:

SI, Pages 26-28, Supplementary Figures 25-27:

The quality of isotope patterns has been improved. A mis-labeling in Supplementary Figure 26 has also been corrected.

Comments by Reviewer #2:

Thiolate-protected gold clusters are currently in the focus of research. Such clusters can serve as model systems to clarify fundamental issues concerning the nature of nanoscale objects and their properties. The present contribution aims at getting fundamental insight on the growth mechanism of gold nanoclusters. This topic is a very important one, as the better understanding could lead to a more rational design of synthesis strategies. In this respect the topic is very appealing and of general interest to many researchers. Despite this I have to confess that I did not learn much new from reading the manuscript. For sure there are some nice points but I definitely expected more. What is demonstrated is that one stable cluster is transformed into another one (, which has been shown before) and that many “intermediate” species are observed underway. A lot remains speculative as already indicated by the language used by the authors (e.g “...which indicates that the size growth most probably occurs via a 2 e-hopping mechanism.”). Furthermore, the importance of the work is diminished by the fact that the reduction was performed by CO, which is a quite special case and there remains always the question whether the growth mechanism would be the same for other, more commonly used reducing agents. For these reasons my feeling is that the physical insight that can really be drawn from the work does not meet the high expectations that the reader has concerning manuscripts published in Nature Communications.

Reply: We're glad that the reviewer agrees with the importance of our research topic. Indeed, revealing the size growth mechanism of nanoclusters (NCs) is vital not only to rational customization of functional NCs for applied research, but also to understanding the transition from molecular state to plasmonic state in noble metal chemistry (*Nat. Commun.* **2016**, 7, 12809; *Nat. Commun.* **2016**, 7, 13240; *Science* **2016**, 354, 1580; *J. Am. Chem. Soc.* **2014**, 137, 1206). We might not have articulated well the new insights revealed in the present work in our initial submission. Through the summary below, we hope to convey the key new physicochemical insights into the NP/NC growth mechanism exemplified in this work.

First, this is the **first** report advancing the mechanistic study of heterogeneous-nucleation-growth of NPs/NCs at the **molecular-level resolution**. As a common practice in organic, inorganic and biological chemistry, reaction pathways are deduced by spectrometrically identifying and tracking the reaction intermediates (Espenson, J. H. *Chemical Kinetics and Reaction Mechanisms*; 2nd ed.; McGraw-Hill, 1995; *Nature* **1990**, 348, 89; *Nature* **2012**, 492, 138; *Nat. Commun.* **2016**, 7, 12223). The reliability of as-deduced reaction mechanism is largely dictated by the accuracy and resolution of analytical tools used for “visualizing” the intermediates. In this work, we adopted such delicate strategy from molecular chemistry, and performed mechanistic study on a model size growth reaction of NCs, $[\text{Au}_{25}(\text{SR})_{18}]^- \rightarrow [\text{Au}_{44}(\text{SR})_{26}]^{2-}$, by tracking **the largest number of intermediate species** (35 in total) through the reaction course **at unprecedented atomic resolution**. This was made possible by combined capabilities of producing seed and product clusters at molecular purity, appropriately slowing down the growth kinetics, and capture of intermediate species by high-resolution mass spectrometry. By doing so, we are not only able to construct each composite reaction map for the chosen model reaction, but also able to answer several fundamental questions of broad interest, like the driving force (boosting of valence electron), initiation (complex formation of seed cluster with reductive species), and size evolution patterns (LaMer-like and aggregative growth) of particle growth. We also exemplified in this manuscript that **the knowledge of the growth mechanism allowed one to rationally steer the synthesis to produce intermediate species of current interest** (e.g., $[\text{Au}_{38}(\text{SR})_{24}]^0$). The successful production of intermediate

$[\text{Au}_{38}(\text{SR})_{24}]^0$ (**first report** on water soluble atomically precise Au_{38} NCs) in turn supports the validity of proposed size growth mechanism.

Second, the fundamental insights into growth mechanism revealed by this study are useful to a diversity of fields, including but certainly not limited to CO-mediated synthesis of metal NPs/NCs. Recent advances in noble metal chemistry have suggested that the two-electron ($2 e^-$) reduction mechanism could be found in other reduction system rather than CO reduction. In particular, Zhu and co-workers have demonstrated that the electrochemical conversion of a silver based quantum dot $[\text{Ag}_{62}\text{S}_{13}(\text{SR})_{32}]^{4+}$ into a silver based nanocluster $[\text{Ag}_{62}\text{S}_{12}(\text{SR})_{32}]^{2+}$ follows a similar $2 e^-$ reduction mechanism (*Chem. Mater.* **2016**, *28*, 7905). In a series of separate work, Aikens and co-worker investigated reduction mechanism of Au(I) precursors via density functional theory (DFT) calculation, and suggested that the Au(0)-Au(0) bonding formation is preferentially occurred via a $2 e^-$ reduction mechanism especially in hydride-reduction approaches (*J. Phys. Chem. Lett.* **2011**, *2*, 990; *J. Phys. Chem. A* **2015**, *119*, 889). The common observation of $2 e^-$ reduction phenomenon in diverse reduction systems implies that such $2 e^-$ hopping feature is likely sourced from superior stability of intermediate NC species with even-numbered valence electron counts over those with odd-numbered counts. Although such implication needs more experimental and/or theoretical support, it is safe to conclude that the $2 e^-$ hopping mechanism is not specific to the CO reduction method. Despite the $2 e^-$ hopping mechanism, this work also mirrors a couple of principles governing the growth of Au NPs/NCs. The most noticeable one is the determinative role of Au(I)-SR complexes in nucleation and growth of Au NPs/NCs. In particular, a fast reduction of reactive Au(I)-SR complexes in Stage 0 could lead to formation of stable $[\text{Au}_{25}(\text{SR})_{18}]^-$ NCs (nucleation), and a further size growth in Stage I should depend on Au_{25} -mediated reduction of inert Au(I)-SR complexes. In addition, the detailed size growth pattern in Stage I is also a combined result of remaining Au(I)-SR complexes and their fitness to the growing Au(0) core. For example, the remaining long Au(I)-SR complex (e.g., $[\text{Au}_{10}(\text{SR})_{10}]^0$) could lead to a size over-growth into $[\text{Au}_{53}(\text{SR})_{41}]^0$, and the preference of short Au(I)-SR motifs by the shrinking curvature of growing Au(0) core could then drive the release of long Au(I)-SR complexes, ultimately giving rise to $[\text{Au}_{44}(\text{SR})_{26}]^{2-}$. Therefore, we believe the present work will attract broad interest from a variety of research communities, including metal-organic complex research, NP/NC research, noble metal chemistry, mechanistic research, and supramolecular chemistry communities.

Last but not least, we are sorry that we didn't articulate well in some terminology in our first submission. We have refined our writing in this revision to improve the preciseness of our phrasing, neither overstating nor understating our findings. Of additional note, the proposed mechanisms based on our time dependent UV-vis and ESI-MS data have also been well rationalized by documented theories and/or independent control experiments. For example, the speculation of Au_{25} -mediated size growth was experimentally supported by successful capture of $[\text{Au}_{25}(\text{SR})_{18}\text{CO}]^-$ intermediate (Figure 6, **first successful attempt** in capturing the cluster-reductant adducts).

Therefore, we believe that current revision of our manuscript (with better conveyed scientific contents) will be appealed to heterogeneous readership of *Nature Communications*.

Revisions:

Page 11, Line 14 – Page 12, Line 3:

“More importantly, each identified species in the ESI-MS spectra carries an even number of valence electrons ($N^* = 0, 2, 6, 8, 10, 12, 14, 16, 18, \text{ and } 20$), which suggests that the clusters grow by 2 e-hopping in size.”

Some more specific points:

1) *The quality of the figures is not sufficient (e.g. in Figure 1 the numbers in the spectra cannot be read), but this maybe a problem of reproduction / resolution.*

Reply: The quality of Figures (Figures 1, 2, 5 and 7) has been improved by enlarging the font size and increasing the image resolution.

Revisions:

Figures 1, 2, 5 and 7:

The font size and resolution have been optimized.

2) *In Figure 2 the spectra are definitely too small, in c) the lines are not distinguishable at all.*

Reply: The quality of Figure 2 has been improved. We would also like to point out that Figure 2c depicts the UV-vis absorption spectra in the size-focusing stage, where most dominant species became $[\text{Au}_{44}(\text{SR})_{26}]^{2-}$. Therefore, the spectra in Figure 2c are expected to be largely superimposable, which may be the source of the confusion.

Revisions:

Page 8, Figure 2:

The size and resolution have been optimized.

3) *The authors start with Au_{25} and then reduce more Au-SR to form further Au_{25} . Is the initial Au_{25} present important in this step? If the reactions starts from just the Au-SR complex is Au_{25} formed as well?*

Reply: Thank you for this insightful comment. Indeed, more $[\text{Au}_{25}(\text{SR})_{18}]^-$ NCs will be formed in the pre-growth stage by CO reduction of reactive Au(I)-SR complexes. Further size growth from $[\text{Au}_{25}(\text{SR})_{18}]^-$ would not occur until those reactive Au(I)-SR complex species have been exhausted (Figures 2 and 3). This observation is somehow different from seed-mediated growth of plasmonic NPs ($> 3 \text{ nm}$), where heterogeneous nucleation is always favored over homogenous nucleation. This discrepancy should be attributed to the discrete stable size at sub-2-nm range and particularly low formation energy of $[\text{Au}_{25}(\text{SR})_{18}]^-$. The insightful comment from the reviewer has encouraged us to explore the dosage effect of starting $[\text{Au}_{25}(\text{SR})_{18}]^-$ NCs. As can be seen in Figure CL-3, a “sweet spot” in terms of $[\text{Au}_{25}(\text{SR})_{18}]^-$ dosage ($< 0.5 \text{ mL}$) could be seen, where monodispersed $[\text{Au}_{44}(\text{SR})_{26}]^{2-}$ could be formed. Further elevating the dosage (1 and 2 mL) would however diversify the cluster size and shift the size distribution downwards. A good example could be found at a dosage of 2 mL, where residual $[\text{Au}_{25}(\text{SR})_{18}]^-$, $[\text{Au}_{38}(\text{SR})_{24}]^0$ and $[\text{Au}_{44}(\text{SR})_{26}]^{2-}$ were formed. This is in good accordance to as-proposed

seed-mediated growth mechanism, where limited Au(I)-SR complexes are not able to drive the size growth reaction of $[\text{Au}_{25}(\text{SR})_{18}]^- \rightarrow [\text{Au}_{44}(\text{SR})_{26}]^{2-}$ to its completion, leaving survival of some stable intermediates (e.g., $[\text{Au}_{38}(\text{SR})_{24}]^0$).

Figure CL-3. Electrospray ionization mass spectra of nanoclusters synthesized at varied dosage of $[\text{Au}_{25}(\text{SR})_{18}]^-$ NCs ($[\text{Au}] = 10 \text{ mM}$). The dotted lines serve as visual guides of $[\text{Au}_{25}(\text{SR})_{18}]^-$ (red), $[\text{Au}_{38}(\text{SR})_{24}]^0$ (green), and $[\text{Au}_{44}(\text{SR})_{26}]^{2-}$ (blue).

It should also be pointed out that despite the dual sources of seed $[\text{Au}_{25}(\text{SR})_{18}]^-$ NCs, the unvaried focus of this study is revealing governing chemistry of size growth reaction from one stable cluster (e.g., $[\text{Au}_{25}(\text{SR})_{18}]^-$) to another (e.g., $[\text{Au}_{44}(\text{SR})_{26}]^{2-}$) at an unprecedented molecular level. This was accomplished by mapping out detailed composite reactions occurring in Stages I (i.e., Au₂₅-assisted size growth) and II (i.e., thermodynamically controlled size focusing). By doing so, we are able to identify the driving force, initiator, as well as detailed size evolution patterns of such reactions, which are believed interesting to heterogeneous readership of *Nature Communications*. By contrast, formation mechanism of $[\text{Au}_{25}(\text{SR})_{18}]^-$ NCs was less touched on this work, as the detailed formation reaction kit of $[\text{Au}_{25}(\text{SR})_{18}]^-$ could be found in our previous publication (*J. Am. Chem. Soc.* **2014**, *136*, 10577).

4) $\text{Au}_4(\text{SR})_4$ is often observed as a fragment in mass spectrometry. Can the authors comment on that.

Reply: Thank you for this insightful comment. It is true that $\text{Au}_4(\text{SR})_4$ is a common fragment of $[\text{Au}_{25}(\text{SR})_{18}]^-$ in mass spectrometry analysis, although such motif (a cyclic tetramer) is not observed in the crystal structure of $[\text{Au}_{25}(\text{SR})_{18}]^-$ or any other $\text{Au}_n(\text{SR})_m$ NCs. To gain insight into the origin of the

$\text{Au}_4(\text{SR})_4$ fragment, we conducted tandem MS (MS/MS) analysis on $[\text{Au}_{25}(\text{SR})_{18}]^-$ NCs, whose structure consists of a center-occupied icosahedral Au_{13} core, capped by six dimeric $\text{Au}_2(\text{SR})_3$ motifs (Figure CL-4a). Tandem MS relies heavily on fragmentation behavior of target molecules under variable collision energy, and has been demonstrated useful for identification and structure analysis of biomolecules and molecular-like clusters (de Hoffmann, E.; Stroobant, V. *Mass Spectrometry: Principles and Applications*; 3rd ed.; Wiley: England, 2007; *J. Am. Chem. Soc.* **2009**, *131*, 13844; *J. Phys. Chem. Lett.* **2014**, *5*, 3757). Monodispersed $[\text{Au}_{25}(\text{SR})_{18} - 2 \text{H}]^{3-}$ ions were selected in 1st round of MS (MS-1), and subjected to fragmentation examination in 2nd round of MS (MS-2). The fragmentation spectra recorded in MS-2 are present in Figure CL-4b. As can be seen, with increasing collision energy up to 30 eV, the intensity of parent $[\text{Au}_{25}(\text{SR})_{18} - 2 \text{H}]^{3-}$ ions decreases while the intensities of fragment ions (i.e., $[\text{Au}_{21}(\text{SR})_{14} - \text{H}]^{2-}$, $[\text{Au}_{22}(\text{SR})_{16} - 2 \text{H}]^{2-}$, and $[\text{Au}_{23}(\text{SR})_{17} - 2 \text{H}]^{2-}$) increase. It is worth noting that these fragment ions are formed by apparent dissociation of single-negatively charged $[\text{Au}_4(\text{SR})_4 - \text{H}]^-$, $[\text{Au}_3(\text{SR})_2]^-$ and $[\text{Au}_2(\text{SR})_1]^-$ modules from the parent ions, respectively. Given the abundance sequence of $[\text{Au}_{23}(\text{SR})_{17} - 2 \text{H}]^{2-} > [\text{Au}_{22}(\text{SR})_{16} - 2 \text{H}]^{2-} > [\text{Au}_{21}(\text{SR})_{14} - \text{H}]^{2-}$ (30 eV panel, Figure CL-4b), we can conclude that the fragmentation of $[\text{Au}_{25}(\text{SR})_{18}]^-$ NCs occurred via dissociation of $\text{Au}_2(\text{SR})_1$, $\text{Au}_3(\text{SR})_2$ and $\text{Au}_4(\text{SR})_4$ complexes, and the preference sequence of dissociation is $\text{Au}_2(\text{SR})_1 > \text{Au}_3(\text{SR})_2 > \text{Au}_4(\text{SR})_4$. Further elevating the collision energy to 40 eV could lead to more severe fragmentation, diminishing the intensity of fragment ions.

[Redacted]

Despite unambiguously identifying $\text{Au}_4(\text{SR})_4$ as a common departing module in MS fragmentation, the detailed departing mechanism remains unclear in current stage of research. However, based on the present tandem MS results, the known crystal structure and other documented structure and reactivity

explorations of $[\text{Au}_{25}(\text{SR})_{18}]^-$, we may be able to construct plausible fragmentation pathways for $[\text{Au}_{25}(\text{SR})_{18}]^-$. Recent achievements in ligand exchange mechanism of $[\text{Au}_{25}(\text{SR})_{18}]^-$ or $[\text{Au}_{24}\text{Pd}(\text{SR})_{18}]^-$ (*J. Phys. Chem. C* **2015**, *119*, 20179; *ACS Nano* **2015**, *9*, 9347; *J. Phys. Chem. C* **2016**, *120*, 25861) suggest the most vulnerable bond as Au2-S1 or symmetric equivalent Au3-S3 (Figure CL-4a, only one symmetrically equal isomer will be described hereafter for clarity) in the frame of $\text{M}_{25}\text{S}_{18}$ (M denotes metal atom). Cleavage of both bonds in the same $\text{Au}_2(\text{SR})_3$ motif results in dissociation of $\text{Au}_2(\text{SR})_1$ module, corresponding to the most prominent fragmentation pathway of $[\text{Au}_{25}(\text{SR})_{18}]^-$ mirrored by our MS/MS analysis. As the protecting motifs of $[\text{Au}_{25}(\text{SR})_{18}]^-$ are uniform $\text{Au}_2(\text{SR})_3$, dissociation of $\text{Au}_3(\text{SR})_2$ module should then involve Au atoms from Au_{13} core. Therefore, the dissociated $\text{Au}_3(\text{SR})_2$ are most probably Au1-S1-Au2-S2-Au3. In sharp contrast to the aforementioned two fragmentation pathways, the dissociation of $\text{Au}_4(\text{SR})_4$ most probably depends on a surface motif deformation-reformation mechanism. Favored cleavage of bond Au2'-S1' in neighbored $\text{Au}_2(\text{SR})_3$ protecting motif would give rise to dangling S1', which could bond to Au1. Subsequent dissociation of S1'-Au1-S1-Au2-S2-Au3-S3-Au4 could result in fragment $\text{Au}_{21}(\text{SR})_{14}$. It should be noted that the structure rearrangement most probably occurs in both the departed small motifs (e.g., $\text{Au}_4(\text{SR})_4$) and the residual fragments (e.g., $\text{Au}_{21}(\text{SR})_{14}$) for stability concern, before they could be captured by MS. Tremendous experimental effort is still needed to fully reveal the fragmentation habit of $[\text{Au}_{25}(\text{SR})_{18}]^-$ and other metal NCs, which are out of scope of current study and could be a good topic for future work.

5) *The authors bubbled CO through the solution to initiate the reaction, then the reaction proceeded in an “air tight” container. This means that initially there was a lot of CO and maybe at a later stage there was only little or none. Can the authors comment on that?*

Reply: Due to continuous consumption with size growth reaction proceeding, a declining concentration of CO is expected in the reaction solution. Such lowered availability of CO is in good accordance with the decayed reducing power required in the proposed three-stage size growth mechanism. Specifically, the strongest reducing power at the beginning of size growth reaction first prompts fast accumulation (<1 h) of Au_{25} in Stage 0. The significant consumption of CO and reactive Au(I)-SR complexes in Stage 0 then allows the reaction to proceed to Stage I with a decreased reducing power, leading to a slowed-down kinetics featured by an elongated duration of ~2 d. It should be pointed out that a dominant population of $[\text{Au}_{44}(\text{SR})_{26}]^{2-}$ has been developed by the end of Stage I (>85%, according to Figure 4), suggesting most CO-consuming reduction reactions are accommodated in Stages 0 and I. In the subsequent Stage II, the remaining CO will be employed to fine-shaping residual NC species with lower valence electron counts (e.g., 14 e-, 16 e- and 18 e-) into $[\text{Au}_{44}(\text{SR})_{26}]^{2-}$. The extraordinarily long duration (~4 d) of this stage should be (at least partially) attributed to the limited availability of CO. It should also be noted that CO-free isoelectronic etching reaction (e.g., $[\text{Au}_{46}(\text{SR})_{27}]^- + 2 \text{SR}^- \rightarrow [\text{Au}_{44}(\text{SR})_{26}]^{2-} + [\text{Au}_2(\text{SR})_3]$) becomes dominant in Stage II, which agrees well with the reduced demand of CO in this stage.

Comments by Reviewer #3:

The authors are studying growth mechanisms of gold-thiolate nanoparticles. They have an approach that uses CO reduction, which occurs slowly enough that they can detect intermediates using ESI-MS. This approach is not used often enough in the community, and definitely has the ability to provide insights into growth mechanisms. There are a number of places where I think the manuscript needs to be improved before being considered ready for publication. The manuscript seems quite long, and I am not sure if Nature Communications is the correct place for it. Nature Chemistry or JACS might be more appropriate.

Reply: We are greatly encouraged by the Reviewer's positive comments in terms of significance and technical innovation of our work. Indeed, our capability of revealing the size growth mechanism of gold nanoparticles at molecular level should be intrinsically attributed to a slow reaction kinetics, which is made possible by the mild gaseous reductant CO and then allow capture of numerous important intermediates by high resolution ESI-MS. We also appreciate the useful and comprehensive comments/suggestions by the reviewer, which have spurred improvements in both readability and scientific content of our manuscript. These comments/suggestions have been taken into careful consideration and a point-to-point response could be found in the coming paragraphs. Regarding the length of manuscript, this revision of our manuscript is within the length limit of *Nature Communications* (up to 5000 words for Introduction, Results and Discussion), and we are ready to move some auxiliary contents (e.g., some experimental details) into Supplementary Information if necessary.

Introduction

1) The authors only cite a previous study by their group about developing a microscopic understanding of growth mechanisms, which might suggest to the reader that only one study has been performed in the past. The authors should include some of the seminal work by the Lennox and Aikens groups that have aided our understanding of microscopic growth mechanisms, including studies on two electron reduction for converting Au(I)SR complexes.

Reply: Thank you for the good suggestion. We have cited and briefly discussed milestone works on the electronic structure and growth mechanism of Au nanoparticles by Aikens, Lennox and other groups (*Chem. Rev.* **2015**, *115*, 6112; *J. Am. Chem. Soc.* **2012**, *134*, 12590; *J. Phys. Chem. Lett.* **2011**, *2*, 990; *J. Phys. Chem. A* **119**, 889-895; *J. Am. Chem. Soc.* **2010**, *132*, 9582; *Langmuir* **2010**, *26*, 13650) in the revised manuscript.

Revision:

Page 4, Lines 16-19:

“Several recent studies also successfully revealed the homogenous nucleation mechanism of Au NPs both theoretically and experimentally, suggesting the importance of chemical states and reduction pathways of Au(I) precursors in Au NP synthesis³⁹⁻⁴².”

Results

2) Numbers in Figure 1 are not readable even when I zoom in on my computer. Resolution in Figure 2 is even worse. Figure 5 needs improvement as well. Both size and resolution need improvement.

Reply: We have maximized the readability of as-mentioned (Figures 1, 2 and 5) and other (Figure 7) figures by optimizing their font size and resolution.

Revisions:

Figures 1, 2, 5 and 7:

The font size and resolution have been optimized.

3) *I would think that the authors would see isosbestic points in their spectra in Figure 2b (and possibly 2c). Do they need to reconsider spectral normalization? (How are the UV-Vis spectra normalized in this work?)*

Reply: We did not observe isosbestic points in our time-evolution UV-vis spectra reported in either Figure 2b or 2c. This may be due to existence of intermediate NC species and their time-dependent population during the size growth reaction. Therefore, the UV-vis spectrometry records absorbance from not only $[\text{Au}_{25}(\text{SR})_{18}]^-$ and $[\text{Au}_{44}(\text{SR})_{26}]^{2-}$, but also other-sized intermediate NCs. We did not normalize UV-vis spectra according to absorbance at a specific wavelength. All UV-vis spectra are present as they were originally recorded.

4) *Rephrase the line “The corresponding species labeled are formulated in Table 1”, as it is not clear what is meant by this sentence.*

Reply: Sorry for the confusion and we have rephrased corresponding line as per suggested.

Revisions:

Page 10, Caption of Figure 3:

“The detailed formula of Au(I)-SR complexes or cluster intermediates identified in mass spectrometry are listed in Table 1.”

5) *Structure “9” seems to be absent in Figure 3, but is listed in Table 1 (and it is later stated in the text that the 35 structures in Table 1 were observed). On the other hand, $[\text{Au}_{12}(\text{SR})_{12}]^0$ is left out of Table 1 (which may be because it does not appear in Figure 3). It is surprising that all of the $\text{Au}_n(\text{SR})_n$ appear in Table 1 except for $n=12$. Also, many of the numbered species in Table 1 are listed as neutral, but they show up in the ESI-MS spectrum. It is only clear much later in the paper when the authors refer to supplementary figures 1-33 that the formulas observed in the ESI-MS are not exactly those listed in the table (varying by H^+ , Na^+ , etc.). It would be good to mention the supplementary figures earlier (and probably it would be good to make it clear that the charges are not the same as those listed in the table). There is some color coding of structure numbers in Figure 3 and Table 1, but it is not explained in either caption. Structure 22 moves in the fourth line down. I think this might be an error. Anyhow, this figure and table need some work.*

Reply: We appreciate the good suggestions of the reviewer to improve the readability of Figure 3 and Table 1. First, as suggested by the reviewer, Supplementary Figures 1-33 were called up in earlier part of main text for an easy reading. Also, the deduction methods for the net charge of clusters have been included as a footnote of Table 1, while the color code of intermediate species have been clearly indicated in the caption of Figure 3. The misleading labeling of species 22 in the 4th panel (40 min) of Figure 3b has been removed, and the labeling of species 9 has been made easier to identify in the 1st panel (0 min) of the same figure. With respect to “missing” species $[\text{Au}_{12}(\text{SR})_{12}]^0$, it does absent in our ESI-MS spectra. This may be due to its reduced stability under a particular solution circumstance employed in this study or susceptibility towards the ionization process. At the current stage of research, a conclusive reason for such missing species is unclear. However, the effects of Au(I)-SR complexes on the size and formation pathway of Au NCs have already gained our attention, and they are under exploration in a separate work.

Revisions:

Page 10, Figure 3:

Ambiguity regarding labeling of species 9 and 22 has been cleared.

Page 10, Caption of Figure 3:

“For ease of identification, the starting cluster $[\text{Au}_{25}(\text{SR})_{18}]^-$, product cluster $[\text{Au}_{44}(\text{SR})_{26}]^{2-}$, as well as an important intermediate cluster $[\text{Au}_{38}(\text{SR})_{24}]^0$ are highlighted in magenta, blue, and olive, respectively.”

Page 11, Footnote of Table 1:

“* All clusters were captured as anions by the form of $[\text{Au}_n(\text{SR})_m + x \text{Na} - y \text{H}]^{q'}$ ($q' < 0$) in ESI-MS analysis (negative ion mode). The net charge of clusters, i.e., q in $[\text{Au}_n(\text{SR})_m]^q$ was deduced via the equation $q = q' - (x - y)$.”

Page 11, Lines 7 – 9:

“Zoom-in ESI-MS spectra and the isotope patterns of all these species are included in Figs. 1d and 1e, and Supplementary Figs. 1-33 (Section S1, Supplementary Information, SI).”

6) It would be good to have ESI-MS of the reaction mixture BEFORE CO addition (when it is just Au(I)-SR complexes prepared as stated on p. 7). The CO reactions may be so fast that it is hard to know what to attribute to the initial reaction mixture and what occurs within a very short time (minutes) after CO is added. For example, the Au₂₅ system must grow very quickly after CO is added (according to the text) without intermediates with Au_n ($n > \sim 15$). It seems likely that some intermediates are too reactive to be observed. However, I don't think the authors should claim that they have seen “all” intermediates since they may have missed some (just because they cannot be observed); they point this out on p. 14, but it is at odds with their abstract (which says they have seen all steps). The number of intermediates they have seen is great, though!

Reply: We are glad to know that our efforts were appreciated by the reviewer. We also agree with the reviewer that some intermediate species may be too reactive to be captured by ESI-MS. This is especially true for intermediate species in the pre-growth stage (Stage 0), where kinetically

accumulation of Au₂₅ completed within 1 h. In a previous study (*J. Am. Chem. Soc.* **2014**, *136*, 10577), we tracked the formation process of [Au₂₅(SR)₁₈]⁻ in a time window of 72 h, which is distinctly longer than that in present study (1 h). This is because we have to apply a stronger reducing power to initiate further size growth from [Au₂₅(SR)₁₈]⁻ to [Au₄₄(SR)₂₆]²⁻ in this work. The stronger reducing power readily fuels a faster reduction kinetics, in which some reactive intermediates could become “invisible” in the ESI-MS analysis. A good example is 4 e⁻ NC species, which is absent in ESI-MS spectra of the present study. We hence re-examined the preciseness of our statement in Abstract as well as in main text, and any overstatement as that spotted by the reviewer has been corrected.

We have also compared the ESI-MS spectra of sole Au(I)-SR complex, [Au₂₅(SR)₁₈]⁻ and a mixture of both in Supplementary Figure 34. It suggests that mixing Au(I)-SR complex and [Au₂₅(SR)₁₈]⁻ would not lead to any chemical reaction. We are thus able to conclude that the Au(I)-SR complex species formed by reacting HAuCl₄ with thiol in this study are [Au_n(SR)_n]⁰ or [Au_n(SR)_{n+1}]⁻ with $n \leq 14$ (Supplementary Table 1).

Revisions:

Page 2, Lines 6-8:

“By systematically investigating time-dependent abundance of totally 35 intermediate species, we map out relevant step reactions in a model size growth reaction from molecularly pure Au₂₅ to Au₄₄ nanoparticles.”

Page 5, Lines 5-7:

“By tracking identifiable intermediate species in the entire course of size growth reaction, we are able to propose a three-stage size hopping mechanism for the seed-mediated formation of [Au₄₄(SR)₂₆]²⁻.”

Page 5, Lines 9-12:

“A detailed investigation on the balanced reactions of identified intermediate species further reveals that the growth of Au NCs is driven by boosting of valence electron count either via a monotonic size growth or a volcano-shaped size-evolution pathway.”

Page 11, Lines 3-4:

“To achieve this, we used ESI-MS to identify stable intermediates during the size conversion.”

SI, Pages 35-36, Supplementary Figure 34 and Supplementary Table 1:

Detailed formula of Au(I)-SR complex species have been included.

7) Observation of even number of valence electrons could imply a 2 electron mechanism, or that odd electron species are very reactive (and thus not observed). It would be good if the authors mention the last point, even if they lean toward the 2 electron mechanism. Moreover, I would not attribute the observation to a “network of superatoms”.

Reply: It is true that the 2 e⁻ hopping observation may be a result of vastly different stabilities of NC species with even and odd valence electron counts. It is widely accepted that the stability of intermediate

species is a determinative factor for reaction pathway. The poor stability of cluster species with odd-numbered valence electron counts could then make reaction pathways involving these species energetically unfavorable, giving rise to a 2 e⁻ hopping mechanism. Due to unknown structures of most intermediates species, we share a similar view with the reviewer that the 2 e⁻ hopping mechanism is better to be attributed to stability difference of even and odd electron species, rather than “a network of superatom”, and we have revised our discussion accordingly.

Revisions:

Page 12, Lines 6-9:

“It should be pointed out that such 2 e⁻ hopping mechanism has also been observed in other reduction systems, indicating that the root cause of the 2 e⁻ hopping mechanism is the relatively good stability of NC species with even-numbered valence electron count^{40, 45, 46}.”

8) In SI, p. 39, the authors mention a proposed decarboxylation of Au₁₀(SR)₁₀COO, but the product they mention has also lost two SR groups. This is not discussed. Things listed on p. 39 are very hypothetical (but it is nice that the authors are thinking about these possibilities). But, they should state that the reactions involved in the size evolution into Au₂₅(SR)₁₈⁻ “may include” (not “also include”) ... and that it is a “proposed” size evolution mechanism (main text, p. 14). Be careful not to overstate your findings if the mechanism is very much a hypothesis.

Reply: We are sorry that we didn't articulate well in some terms, which have been re-examined and corrected with maximum care in this revision. The carboxylation-decarboxylation mechanism is originally proposed in our previous study (*J. Am. Chem. Soc.* **2014**, *136*, 10577), where homogenous-nucleation-growth of [Au₂₅(SR)₁₈]⁻ was investigated, and experimentally supported by presence of stoichiometric amount of CO₃²⁻ in total inorganic carbon analysis. Since the detailed size evolution mechanism is discussed elsewhere (*J. Am. Chem. Soc.* **2014**, *136*, 10577), only brief descriptions of reactions involved in such size evolution process are present in the Supplementary Information. To minimize any possible confusion, we included more details about the proposed carboxylation-decarboxylation mechanism (Supplementary Figure 35 and corresponding explanative lines in Supplementary Notes for Stage 0).

Revisions:

SI, Page 37, Supplementary Figure 35:

A supplementary figure schematically illustrating plausible pathway for the carboxylation-decarboxylation reduction of Au(I)-SR complexes has been included.

SI, Page 41, Lines 7-13

“As shown in Supplementary Note 1, [Au₁₀(SR)₁₀]⁰ would first associate with a CO to form [Au₁₀(SR)₁₀CO]⁰ adduct, which could be converted to [Au₁₀(SR)₁₀COOH]⁻ via a typical Hieber base reaction at alkaline condition (pH = 13.0). Decarboxylation of such adduct then transfers 2 e⁻ to neighbored Au(I) centers and simultaneously releases two free SR ligands, giving rise to [Au₁₀(SR)₈]⁰ which contains two Au(0) centers. A plausible reaction pathway for such carboxylation-decarboxylation process can be seen in Supplementary Fig. 35.”

SI, Page 42, Supplementary Note 1:

“ ...

Hieber Base Reaction: $[\text{Au}_{10}(\text{SR})_{10}\text{CO}]^0 + \text{OH}^- \rightarrow [\text{Au}_{10}(\text{SR})_{10}\text{COOH}]^-$

Decarboxylation: $[\text{Au}_{10}(\text{SR})_{10}\text{COOH}]^- + 3 \text{OH}^- \rightarrow [\text{Au}_{10}(\text{SR})_8]^0 + \text{CO}_3^{2-} + 2 \text{H}_2\text{O} + 2 \text{SR}^-$

...”

Page 15, Lines 20 – 25

“In addition to the aforementioned reduction-growth, reactions involved in the size evolution into $[\text{Au}_{25}(\text{SR})_{18}]^-$ may also include isoelectronic addition, comproportionation, and isoelectronic etching, which are in good agreement with our previous findings³⁰. More details about the proposed size evolution mechanism are included in the SI (Supplementary Fig. 35 in Section S1, and Supplementary Notes 1 and 2 in Section S2).”

9) *Figure 5 shows one option of $\text{Au}_{25}(\text{SR})_{18}^-$ undergoing aggregative growth to $\text{Au}_{35}(\text{SR})_{28}^{3-}$. This would mean addition of $\text{Au}_{10}(\text{SR})_{10}^{2-}$, which was not mentioned as an intermediate. It is eventually discussed on p. 18, but the charge changes and no hypothesis is given.*

Reply: The formation of $[\text{Au}_{35}(\text{SR})_{28}]^{3-}$ species is more likely via a two-step reaction of $[\text{Au}_{25}(\text{SR})_{18}\text{CO}]^-$. A $[\text{Au}_{10}(\text{SR})_{10}]^0$ would first adsorb on $[\text{Au}_{25}(\text{SR})_{18}\text{CO}]^-$, yielding $[\text{Au}_{35}(\text{SR})_{28}\text{CO}]^-$ intermediate. Cleavage of CO from $[\text{Au}_{35}(\text{SR})_{28}\text{CO}]^-$ via carboxylation-decarboxylation process could then transfer 2 e⁻ to the cluster species, giving rise to $[\text{Au}_{35}(\text{SR})_{28}]^{3-}$. More details about the formation process of $[\text{Au}_{35}(\text{SR})_{28}]^{3-}$ has been included.

Revisions:

Page 19, Line 22 – Page 20, Line 1

“Specifically, as depicted in Supplementary Note 4, the $[\text{Au}_{25}(\text{SR})_{18}\text{CO}]^-$ would react with a long Au(I)-SR complex (e.g., $[\text{Au}_{10}(\text{SR})_{10}]^0$), giving rise to $[\text{Au}_{35}(\text{SR})_{28}\text{CO}]^-$ ($N^* = 8$) and eventually evolving into $[\text{Au}_{35}(\text{SR})_{28}]^{3-}$ ($N^* = 10$) via the aforementioned carboxylation-decarboxylation mechanism.”

10) *The solubility of the Au(I)-SR complexes has been questioned and appears to depend on the R group, solvent, and pH. The authors use high pH and water-soluble R groups, which is known to increase the solubility of the Au(I)-SR complexes. However, the things learned under these conditions may not apply to other conditions (especially ones in which the Au(I)-SR complexes are not as soluble). These considerations should be briefly discussed in the manuscript.*

Reply: Thank you for this insightful comment. We agree that the solubility and other physicochemical properties (e.g., size and structure) of Au(I)-SR complexes are dependent on their intrinsic (e.g., R group) and surrounding (e.g., solvent polarity, pH and ionic strength) conditions. The size, structure and solubility of Au(I)-SR complexes would determine their reactivity towards reduction, and ultimately affect the size and monodispersity of product clusters.

Revisions:

Page 14, Lines 13-15

“Of note, the chemical identity and solubility of Au(I)-SR complexes could vary with different intrinsic (e.g., R group) and surrounding (e.g., pH, ionic strength, and solvent polarity) conditions, which can affect the cluster growth mechanism.”

11) *The purpose of Figure 5 is not clear. 5a was mentioned in the text (but not really explained).*

Reply: We used Figure 5 to schematically illustrate the relative reactivity of Au(I)-SR complexes and $[\text{Au}_{25}(\text{SR})_{18}]^-$ NCs towards CO reduction (Figure 5a), and detailed size evolution patterns from $[\text{Au}_{25}(\text{SR})_{18}]^-$ to $[\text{Au}_{44}(\text{SR})_{26}]^{2-}$ (Figure 5b). More descriptive lines have been added to better illustrating Figure 5.

Revisions:

Page 15, Lines 8-13

“As illustrated in Fig. 5a, this preferred reaction is driven by the lower activation energy (E_a) of the reactive Au(I)-SR complex species to form $[\text{Au}_n(\text{SR})_n\text{CO}]^0$ intermediates over CO complexing with $[\text{Au}_{25}(\text{SR})_{18}]^-$ (i.e., formation of $[\text{Au}_{25}(\text{SR})_{18}\text{CO}]^-$ intermediate). The reduction could then occur by transferring 2 e- from CO to the complex species via a carboxylation-decarboxylation mechanism³⁰, generating the NC species with $N^* = 2$.”

Page 18, Line 1 – Page 19, Line 1

“The most intriguing finding in the Au₂₅-mediated size-growth stage is a dual-mode size growth pattern (Fig. 5b), which is inherently dictated by the co-existence of less reactive Au(I)-SR complex (e.g., $[\text{Au}_4(\text{SR})_5]$) and NC (e.g., $[\text{Au}_{18}(\text{SR})_{14}]^{2+}$) species. The upper pathway in Fig. 5b shows a monotonic size growth pattern. This pathway relies on the addition of newly reduced Au(I)-SR complexes on the growing NCs, and thus it could be considered as an analogue of LaMer mechanism.”

Page 19, Lines 18-22

“As shown in Fig. 5b (down pathway), the size evolution follows a detailed pattern of 8 e- (i.e., $[\text{Au}_{25}(\text{SR})_{18}]^-$) \rightarrow 10 e- (i.e., $[\text{Au}_{35}(\text{SR})_{28}]^{3-}$) \rightarrow 12 e- (i.e., $[\text{Au}_{53}(\text{SR})_{41}]^0$) \rightarrow 14 e- (i.e., $[\text{Au}_{40}(\text{SR})_{25}]^+$ and $[\text{Au}_{51}(\text{SR})_{38}]^-$) \rightarrow 16 e- (i.e., $[\text{Au}_{42}(\text{SR})_{25}]^+$) \rightarrow 18 e- (i.e., $[\text{Au}_{43}(\text{SR})_{24}]^+$) \rightarrow 20 e- (i.e., $[\text{Au}_{44}(\text{SR})_{26}]^{2-}$ and $[\text{Au}_{46}(\text{SR})_{27}]^-$).”

12) *p. 16 $[\text{Au}_{18}(\text{SR})_{14}]^{2+}$ is mentioned. How could the positively charged species (especially with 2+) form under these basic reaction conditions? The plausibility of this should be discussed.*

Reply: The net charge (q) of $[\text{Au}_n(\text{SR})_m]^q$ is determined according to the ions recorded in ESI-MS spectrum, $[\text{Au}_n(\text{SR})_m + x \text{Na} - y \text{H}]^q$, via the equation $q = q' - (x - y)$. This does not readily mean the cluster species exist in such intrinsic form (i.e., $[\text{Au}_n(\text{SR})_m]^q$) in solution. By contrast, they are most probably present as anionic form by deprotonation of carboxylic group of SR ligand (*para-*

mercaptobenzoic acid or *p*-MBA in this study). The high stability of clusters in anionic form in solution could be indirectly supported by ESI-MS results, where stable cluster anions could be captured. For example, $[\text{Au}_{18}(\text{SR})_{14}]^{2+}$ could be captured in the anionic form of $[\text{Au}_{18}(\text{SR})_{14} + \text{Na} - 8 \text{H}]^{5-}$ in the ESI-MS (Supplementary Figure 16), suggesting high stability of such cluster anion. We thus tend to believe $[\text{Au}_{18}(\text{SR})_{14}]^{2+}$ should be accommodated in anionic form with negative charges developed by deprotonation of surface carboxylic group in the solution. The deduction method of net charge of cluster has been clearly described in this revision.

Revisions:

Page 11, Footnote of Table 1:

“* All clusters were captured as anions by the form of $[\text{Au}_n(\text{SR})_m + x \text{Na} - y \text{H}]^{q'}$ ($q' < 0$) in ESI-MS analysis (negative ion mode). The net charge of clusters, i.e., q in $[\text{Au}_n(\text{SR})_m]^q$ was deduced via the equation $q = q' - (x - y)$.”

13) p. 16 “*should be attributed*” should be rephrased (“*could be attributed*”), since it is referring to something that is still very much a hypothesis.

Reply: The suggested correction has been made.

Revisions:

Page 17, Lines 15-18:

“Therefore, the formation of $[\text{Au}_{25}(\text{SR})_{18}\text{CO}]^-$ could be attributed to the accommodation of CO in those pocket-like sites, and in turn the strong binding between CO and Au might make CO more susceptible to oxidation.”

14) p. 17 “*is proven*” should be “*is supported*”. Again, don’t overstate your findings.

Reply: Corrected.

Revisions:

Page 19, Lines 11-14:

“Such a bottom-up step-wise (with a pace of 2 e-) growth process from 8 e- to 20 e- NC species is supported by the temporal appearance sequence of the corresponding NC species (Fig. 4).”

15) p. 18 The authors have not explained why the aggregative growth pathway is “volcano-shaped”. I infer that it is because they suggest the particles get bigger before they get smaller, but the term volcano is used throughout chemistry in a variety of contexts, and should be explained.

Reply: Yes, we are referring to a size over-growth of $[\text{Au}_{25}(\text{SR})_{18}]^-$ into $[\text{Au}_{53}(\text{SR})_{41}]^0$ followed by size-down to $[\text{Au}_{44}(\text{SR})_{26}]^{2-}$. Explanation about the term “volcano-shaped” has been added.

Revisions:

Page 19, Lines 16-18:

“In addition to bottom-up LaMer-like growth pathway, Fig. 4 also suggests an alternative volcano-shaped growth pattern, where $[\text{Au}_{25}(\text{SR})_{18}]^-$ first over-grows into $[\text{Au}_{53}(\text{SR})_{41}]^0$ followed by reduction-assisted size-down to $[\text{Au}_{44}(\text{SR})_{26}]^{2-}$.”

16) *The authors could consider doing a kinetic analysis of their system to see if these mechanisms are supported by the available information. This would be difficult (and certainly not expected for the current paper), but it really could help support or refute some of the hypotheses they have that can not currently be supported in detail.*

Reply: Very good suggestion. Our follow-up efforts will be devoted into suggested kinetic analysis. We can foresee both the significance as well as the difficulty for such kinetic analysis.

17) *The authors considered only changing the SR-to-Au ratio or the pH of the solution and said they do not achieve the Au_{38} species. However, the supplementary figure 35 appears to have Au_{38} in it (where only the SR-to-Au ratio is changed). Is this correct?*

Reply: We explored the SR-to-Au ratio effects on the size of clusters in the seed-mediated synthesis. At a ratio of 1 : 1, we were able to produce high purity $[\text{Au}_{44}(\text{SR})_{26}]^{2-}$. However, by solely changing the SR-to-Au ratio (to 2 : 1 or 3 : 1), we could only obtain a mixture of $[\text{Au}_{38}(\text{SR})_{24}]^0$ and $[\text{Au}_{44}(\text{SR})_{26}]^{2-}$. No monodispersed $[\text{Au}_{38}(\text{SR})_{24}]^0$ was synthesized under the reaction conditions quoted in Supplementary Figure 35 (now Supplementary Figure 36).

18) *The TDDFT spectrum for the neutral Au_{44} system should also be presented so that the differences between theory and experiment for this system with this level of theory can be assessed. The spectra presented in Supplementary figure 37 differs a fair bit. The methods section says def2-TZVP, but the SI section says def2-SV(P). Could the authors change one of these?*

Reply: The difference between the simulated and experimental optical absorption spectra for the dianion in Supplementary Figure 37 (Supplementary Figure 38 in this revision) could be attributed to the simplified SR ligand (-SCH₃) used in the time-dependent DFT (TDDFT) calculation. As the reviewer suggested, we also computed the TDDFT spectrum of the neutral cluster and added it in Supplementary Figure 38. One can see that the agreement with the experiment is worse for the simulated neutral cluster, supporting our assignment of 2- charge for the cluster. We have added this discussion in the revised manuscript. In addition, we have corrected the description of the basis sets: both places should be def2-SV(P).

Revisions:

SI, Page 40, Supplementary Figure 38:

The simulated optical absorption spectrum of $[\text{Au}_{44}(\text{SR})_{26}]^0$ has been included.

Page 24, Lines 12-18:

“The time-dependent DFT (TDDFT) simulation at the B3LYP level yielded a spectrum (Supplementary Fig. 38) showing several distinct bands in reasonable agreement with the experiment; the differences between the simulation and the experiment could be due to the simple -SCH₃ used for -SR in our simulation. We also computed the TDDFT spectrum of the neutral $[\text{Au}_{44}(\text{SR})_{26}]^0$ (Supplementary Fig. 38), whose worse agreement with experiment supported the 2-charge state in as-obtained Au₄₄ NCs.”

Page 28, Lines 17-19:

“...the def2-SV(P) basis sets were performed with the quantum chemistry program Turbomole V6.5⁶⁰.”

Overall comment:

The authors should be commended for tackling a difficult problem that has the potential to shed light on growth mechanisms. However, there is currently a tendency to overstate their hypotheses as “mechanisms” without direct support.

In a number of places in the manuscript, there is some awkward phrasing. You may wish to go through and wordsmith the document some more to remove these.

Reply: We are especially encouraged by the reviewer’s positive acknowledgement of our efforts on tackling this long-standing puzzle (growth mechanism of NCs/NPs). We are also grateful to the reviewer for his/her insightful and comprehensive comments/suggestions, which are useful for improving the readability and scientific content of our manuscript. We have improved preciseness of our data interpretation in current revision (revisions not described in previous context could be found at the end of this letter). We have also revised the manuscript in language wise to diminish improper phrasing. We hope current revision is satisfactory to the reviewer.

Additional Revisions:

Page 19, Lines 8-11:

“The as-formed 10 e- $[\text{Au}_{28}(\text{SR})_{21}]^{3-}$ would subsequently evolve into 12 e- (i.e., $[\text{Au}_{33}(\text{SR})_{22}]^-$) → 14 e- (i.e., $[\text{Au}_{37}(\text{SR})_{23}]^0$ and $[\text{Au}_{38}(\text{SR})_{24}]^0$) → 16 e- (i.e., $[\text{Au}_{42}(\text{SR})_{25}]^+$) → 18 e- (i.e., $[\text{Au}_{43}(\text{SR})_{24}]^+$) → 20 e- (i.e., $[\text{Au}_{44}(\text{SR})_{26}]^{2-}$ and $[\text{Au}_{46}(\text{SR})_{27}]^-$) NCs, via the common reduction-growth mechanism.”

Page 21, Lines 1-3:

“Further size evolution of $[\text{Au}_{42}(\text{SR})_{25}]^+$ ($N^* = 16$) into $[\text{Au}_{43}(\text{SR})_{24}]^+$ ($N^* = 18$), $[\text{Au}_{44}(\text{SR})_{26}]^{2-}$ and

[Au₄₆(SR)₂₇]⁻ (*N** = 20) species could be readily accomplished via the typical reduction-growth mechanism.”

Page 22, Lines 4-6:

“The dominant reactions involved in this stage are therefore reduction-growth and isoelectronic etching, as exemplified in Supplementary Note 6.”

Reviewers' comments:

Reviewer #1 (Remarks to the Author):

The newly revised manuscript addresses the referee comments. I recommend the publication of this work in Nature Communications.

Reviewer #2 (Remarks to the Author):

The authors have responded to my questions and made changes to the manuscript accordingly. I think that the manuscript is now improved. The manuscript definitely shows nice results. My major concern is still remaining, namely that there is no real proof for the two-electron hopping mechanism. The apparent absence of clusters with uneven electron counts in the mass spectra does not mean that they do not exist. It likely means that they are unstable and reactive. I think the paper can be published but I feel it would be better for example in JACS.

Reviewer #3 (Remarks to the Author):

Comments on paper:

Overall, the authors have improved the paper significantly from the first version. I am still not sure that it is appropriate for Nature Communications, however.

Coloring in Table 1 does not match the coloring in Figure 3 for the Au₂₅ species (red vs. magenta). Change Au₂₅ coloring in Table 1 for consistency.

In Figure 4, the Au₁₈ species formula appears to have 11 thiolates. Should that be 14 rather than 11? I am not sure if it is just my PDF that looks like that. The blue dotted line may go through the bottom of the 14 and may need to be moved slightly.

I am glad the authors added the line "Of note, the chemical identity and solubility of Au(I)-SR complexes could vary with different intrinsic (e.g., R group) and surrounding (e.g., pH, ionic strength, and solvent polarity) conditions, which can affect the cluster growth mechanism." However, it seems a bit out of place in its current location (on p. 14). Is there a better place to put it (earlier or later) in the text?

I am not really sure about the comment "The formation of minor amounts of these NC species ($N^* > 0$) could be attributed to the enhanced reduction power of thiolate ligands at elevated pH (13.0), which could reduce Au(III) to Au(0) by boosting the oxidation state of S to +4 or +6^{48, 49}." The authors are not really dealing with Au(III) in that reaction mixture (are they?) but instead Au(I)SR (correct?). Or, do you specifically mean from the Au(III) salt? Also, I don't think S actually reaches these oxidation states in the pathways suggested in this paper.

Figure 5a is not a good graph, since many different species are shown on the same graph but are connected with a single solid line, which typically indicates connections between these species in a reaction pathway. Also, it is unclear whether the relative free energies of the Au_n(SR)_n, Au₂₅(SR)₁₈-, Au₃₈(SR)₁₄-, etc. are intended in that way, and, if so, how these are determined. This graph needs to be substantially reworked.

The x-axis of Supplementary Figure 38 should be the same for all species. It is really not clear if -2 or

0 is truly better than the other in comparison with experiment. Another reason for differences in theory and experiment that could be mentioned in the text (in addition to the difference in ligand used) is the functional and basis set chosen.

For supplementary notes 1 and 2, it would probably be best to say "Possible formation reaction pathways..." and "Possible isoelectronic addition..." (or use "Proposed" instead of "Possible")

On p. 26, the sentence starting "On the basis of the understandings" does not read quite right. I'm not sure if it is a simple from/form issue or if more needs to be done with this sentence. It may be possible to reword the Discussion section in general a bit more for higher impact and greater readability.

Comments on response to reviewers document:

p. 2 I am not sure that the overlapping PAGE gel bands can be used as proof that only one cluster exists, since they appear to overlap one another.

p. 3 I am not sure that a discrepancy between experimental and simulated mass spectra should be attributed to LOW signal-to-noise (and it is not clear what "better defined" means to me). However, since this is not discussed in the actual text, it is up to the other reviewer if this point was reasonably explained.

p. 12 Since the spectra are not normalized, this should be stated in the methods section. Were the ESI-MS and UV-vis measurements performed on the same sample for the various times? (see p. 9 of the main text) If so, how could that affect/not affect the results?

Summary:

Overall, the authors have done a nice job of elucidating some of the intermediates that can be involved in nanoparticle growth. The mechanisms are still a bit uncertain, but it will be a while before these can be determined one way or another. It is good that the authors have now phrased things more as suggestions rather than as proven mechanisms. The paper will appeal to specialists in this area, but I am not sure it will appeal to the broader Nature Communications audience. A more specialized journal may still be more appropriate.

Responses to reviewers' comments and description of revisions made

Comments by Reviewer #3:

Comments on paper:

Overall, the authors have improved the paper significantly from the first version. I am still not sure that it is appropriate for Nature Communications, however.

Reply: We are glad that the reviewer finds significant improvements in the revised manuscript. On-demand manipulation of materials at atom-by-atom basis is the holy grail of material science and chemistry. Such dream should be rooted in precise understandings on formation and/or assembly mechanism of functional materials, which has gained considerable fundamental interests in various fields like crystalline and/or non-crystalline metal (*Nature* **2008**, 451, 549; *Nat. Chem.* **2017**, 9, 77; *Nat. Nano.* **2011**, 6, 93; *J. Am. Chem. Soc.* **2012**, 134, 12590; *J. Am. Chem. Soc.* **2010**, 132, 9582; *J. Am. Chem. Soc.* **2014**, 136, 10577), magnetite (*J. Am. Chem. Soc.* **2013**, 135, 2407; *J. Am. Chem. Soc.* **2007**, 129, 12571), and semiconductor (*Nature* **2016**, 531, 317; *Nat. Mater.* **2016**, 15, 1248; *Nat. Commun.* **2017**, 8, 15467; *Nat. Commun.* **2016**, 7, 12223) nanoparticle (NP) research.

Our present work is a demonstrative example of unraveling growth mechanism of inorganic NPs at unprecedented *molecular-level*, based on combined spectrometric and simulated analyses on the largest ever number of intermediate species in seed-mediated growth of Au nanoclusters (NCs). The insights gleaned into the initiation, driving force and detailed size evolution patterns of Au NCs will not only be of interest to the metal NP/NC research community, but also show broad implications in other branches of inorganic materials research, such as metal chemistry, complex chemistry, quantum dots research, and catalysis research. For example, the *first-ever capture* of CO-Au NC adducts (i.e., $[\text{Au}_{25}(\text{SR})_{18}\text{CO}]^-$) at isotope resolution would add to the mechanistic understanding on metal-catalyzed oxidation of CO, which is one of the most important topics in current sustainability research. In addition, this work demonstrated a facile seed-mediated synthetic method capable of producing a series of atomically precise metal NCs (e.g., $[\text{Au}_{25}(\text{SR})_{18}]^-$, $[\text{Au}_{38}(\text{SR})_{24}]^0$ and $[\text{Au}_{44}(\text{SR})_{26}]^{2-}$) in aqueous solution, which could be useful to any fundamental (e.g., size-structure-property evolution) and applied (e.g., healthcare, environment, catalysis, and energy) study using metal NCs. Therefore, we believe this work will be of interest to the wide and heterogeneous readers of *Nature Communications*, and it will rapidly stimulate more studies on synthesis, formation mechanism and practical applications of noble metal and other functional nanomaterials.

1) Coloring in Table 1 does not match the coloring in Figure 3 for the Au_{25} species (red vs. magenta). Change Au_{25} coloring in Table 1 for consistency.

Reply: Thanks for spotting our coloring mistake. We have changed it accordingly.

Revision:

Page 11, Table 1:

The color codes of $[\text{Au}_{25}(\text{SR})_{18}]^-$ and $[\text{Au}_{38}(\text{SR})_{24}]^0$ in Table 1 have been changed (red to magenta for the former, and green to olive for the latter).

2) In Figure 4, the Au₁₈ species formula appears to have 11 thiolates. Should that be 14 rather than 11? I am not sure if it is just my PDF that looks like that. The blue dotted line may go through the bottom of the 14 and may need to be moved slightly.

Reply: We are sorry for this typo, and have corrected it accordingly in this revision. Thank you. We have also re-examined the formula of all other species for a correct and accurate presentation.

Revision:

Page 13, Figure 4:

The formula of [Au₁₈(SR)₁₁]²⁺ has been corrected as [Au₁₈(SR)₁₄]²⁺.

3) I am glad the authors added the line “Of note, the chemical identity and solubility of Au(I)-SR complexes could vary with different intrinsic (e.g., R group) and surrounding (e.g., pH, ionic strength, and solvent polarity) conditions, which can affect the cluster growth mechanism.” However, it seems a bit out of place in its current location (on p. 14). Is there a better place to put it (earlier or later) in the text?

Reply: Thank you again for this good suggestion. We have rephrased and placed the quoted sentences to an earlier part of the same paragraph.

Revision:

Page 14, Lines 5-10:

“Before considering the details of Au(I)-SR complex species in the reaction mixture, it should be noted that the chemical identity and solubility of Au(I)-SR complexes could vary with different intrinsic (e.g., R group) and surrounding (e.g., pH, ionic strength, and solvent polarity) conditions. Based on their various identities and structures, Au(I)-SR complexes species would show varied reactivity towards CO reduction, affecting the cluster growth mechanism.”

4) I am not really sure about the comment “The formation of minor amounts of these NC species ($N^* > 0$) could be attributed to the enhanced reduction power of thiolate ligands at elevated pH (13.0), which could reduce Au(III) to Au(0) by boosting the oxidation state of S to +4 or +6^{48, 49}.” The authors are not really dealing with Au(III) in that reaction mixture (are they?) but instead Au(I)SR (correct?). Or, do you specifically mean from the Au(III) salt? Also, I don’t think S actually reaches these oxidation states in the pathways suggested in this paper.

Reply: Thank you for your insightful comment. We totally agree with the reviewer that CO (upon the addition) is the sole reducing agent that fuels the growth of Au(0) core. The quoted lines serve as an additional remark accounting for the observation of trace amount of Au(0) species in the reaction mixture prior to CO addition. It has been documented that excess Au(III) ions are capable of oxidizing thiols to higher oxidation states (e.g., +4 or +6) in aqueous solution (*Chem. Rev.* **1999**, 99, 2589; *Inorg. Chem.* **1980**, 19, 3198; *J. Chem. Soc., Chem. Commun.* **1981**, 1111; *J. Am. Chem. Soc.* **2012**, 134, 16662), with Au(0) as reduction product. We hence posit that the minor amount of Au(0) species observed prior to CO addition should be formed by such redox reaction between Au(III) salt and thiol.

5) Figure 5a is not a good graph, since many different species are shown on the same graph but are connected with a single solid line, which typically indicates connections between these species in a reaction pathway. Also, it is unclear whether the relative free energies of the $Au_n(SR)_n$, $Au_{25}(SR)_{18}^-$, $Au_{38}(SR)_{24}$, etc. are intended in that way, and, if so, how these are determined. This graph needs to be substantially reworked.

Reply: We are very thankful for your good suggestions, and have made the changes accordingly. Figure 5a is a qualitative illustration of relative free energetics of $[Au_n(SR)_m]^q$ species, giving rise to the observed three stage growth mechanism. To avoid confusion, it has now been redrawn, based on their reactivity and stability in CO-saturated solution (e.g., reactivity: $[Au_n(SR)_n]^0 > [Au_{25}(SR)_{18}]^- > [Au_{38}(SR)_{24}]^0 > [Au_{44}(SR)_{26}]^{2-}$). The non-elementary reaction pathways are now indicated by dotted lines. Necessary legend description has also been included in the caption of Figure 5a.

Revisions:

Page 16, Figure 5 and its caption:

“ $E_{a,Au(I)-SR}$ and $E_{a,Au25}$ denotes the formation energy of $[Au_n(SR)_nCO]^0$ and $[Au_{25}(SR)_{18}CO]^-$, respectively; the relative energies are schematically drawn according to the stability of $[Au_n(SR)_m]^q$ species in CO-saturated solution; the dotted curves indicate non-elementary reaction pathways.”

6) The x-axis of Supplementary Figure 38 should be the same for all species. It is really not clear if -2 or 0 is truly better than the other in comparison with experiment. Another reason for differences in theory and experiment that could be mentioned in the text (in addition to the difference in ligand used) is the functional and basis set chosen.

Reply: Thank you for this good suggestion. As suggested, the spectra in Supplementary Figure 38 have now been re-drawn under the same x-axis scale. For easy comparison, we also highlighted the main absorption peaks of $[Au_{44}(SR)_{26}]^q$ ($q = 0$ or -2) in regions I, II, and III, respectively. As can be seen in the top spectrum, the neutralized cluster charge would split peak II into doublet, distinctively different from the singlet absorption peak in its experimental spectrum. Also, the simulated peaks I and III of $[Au_{44}(SR)_{26}]^{2-}$ exhibit closer peak positions to the experimental ones. We therefore conclude that the simulated spectrum of $[Au_{44}(SR)_{26}]^{2-}$ is a better mimic of the experimental one. The suggested reasons accounting for the differences between simulated and experimental spectra have also been added into the revised manuscript.

Revisions:

Supplementary Information, Page 40, Supplementary Figure 38:

Supplementary Figure 38. Simulated optical absorption spectra of $[\text{Au}_{44}(\text{SR})_{26}]^q$ for $q = 0$ (neutral) and -2 (dianion), in comparison with the experimental spectrum of $[\text{Au}_{44}(\text{SR})_{26}]^{2-}$. Simulation was done at the TDDFT-B3LYP/def2-SV(P) level for $[\text{Au}_{44}(\text{SCH}_3)_{26}]^q$. The main absorption peaks are highlighted in regions I, II and III, respectively, for easy comparison.

Page 24, Lines 14-16:

“... the differences between the simulation and the experimental results could be due to simplified functional and basis set, and different ligands ($-\text{SCH}_3$ used for $-\text{SR}$) used in our simulation.”

7) For supplementary notes 1 and 2, it would probably be best to say “Possible formation reaction pathways...” and “Possible isoelectronic addition...” (or use “Proposed” instead of “Possible”).

Reply: We have made the suggested changes accordingly.

Revisions:

Supplementary Information, Page 42, Supplementary Note 1:

“Proposed formation reaction pathways of $[\text{Au}_{15}(\text{SR})_{13}]^0$ by CO-reduction of $[\text{Au}_{10}(\text{SR})_{10}]^0$.”

Supplementary Information, Page 43, Supplementary Note 2:

“Proposed isoelectronic addition, isoelectronic etching, and comproportionation reactions occurring in formation of $[\text{Au}_{25}(\text{SR})_{18}]^-$.”

8) On p. 26, the sentence starting “On the basis of the understandings” does not read quite right. I’m not sure if it is a simple from/form issue or if more needs to be done with this sentence. It may be possible to reword the Discussion section in general a bit more for higher impact and greater readability.

Reply: We have rephrased the section Discussion (including as-mentioned lines) for a general and impactful readability. We really appreciated the reviewer’s good suggestions on improving the readability of our paper to the diverse readers of *Nature Communications*.

Revisions:

Page 25, Line 21 – Page 26, Line 5:

“With a systematic investigation into the formation and consumption reactions of all 35 Au(I)-SR complex/NC species captured in the ESI-MS spectra, we identified the initiation (i.e., adsorption of CO on $[\text{Au}_{25}(\text{SR})_{18}]^-$), driving force (i.e., 2 e⁻ boosting valence electron counts), and detailed size evolution patterns (i.e., LaMer-like and aggregative growth) for the size growth reactions from $[\text{Au}_{25}(\text{SR})_{18}]^-$ to $[\text{Au}_{44}(\text{SR})_{26}]^{2-}$. Based on these molecular-level insights on cluster/particle growth, we were also able to drive the seed-mediated growth reaction kit to produce intermediate sizes (e.g., molecularly pure $[\text{Au}_{38}(\text{SR})_{24}]^0$.”

Page 26, Lines 9 – 10:

“...which have puzzled the nanoscience and nanomaterials research communities for decades.”

Comments on response to reviewers document:

9) p. 2 I am not sure that the overlapping PAGE gel bands can be used as proof that only one cluster exists, since they appear to overlap one another.

Reply: We have also noted that *p*-mercaptobenzoic acid (*p*-MBA) protected Au NCs exhibited remarkable axial diffusion in the PAGE gel, giving rise to broadened PAGE bands (in comparison to those of peptide-protected Au NCs, *J. Am. Chem. Soc.* **2014**, 136, 1246; *J. Am. Chem. Soc.* **2005**, 127, 5261). Similar axial diffusion induced width broadening was also observed in PAGE bands of *p*-MBA protected Ag NCs (*Nature* **2013**, 501, 399). Through careful comparison of the diffusion fronts of NCs, an ion mobility sequence could be identified as $[\text{Au}_{25}(\textit{p}\text{-MBA})_{18}]^- > [\text{Au}_{38}(\textit{p}\text{-MBA})_{24}]^0 > [\text{Au}_{44}(\textit{p}\text{-MBA})_{26}]^{2-}$ (Figure RL-1), which is in good accordance to their size sequence. It should be pointed out that a high purity of $[\text{Au}_{25}(\textit{p}\text{-MBA})_{18}]^-$, $[\text{Au}_{38}(\textit{p}\text{-MBA})_{24}]^0$ and $[\text{Au}_{44}(\textit{p}\text{-MBA})_{26}]^{2-}$ was supported by their clean ESI-MS spectra (Figures 1 and 7).

Figure RL-1. PAGE results of $[\text{Au}_{25}(\text{SR})_{18}]^{-}$ (left), $[\text{Au}_{38}(\text{SR})_{24}]^0$ (middle) and $[\text{Au}_{44}(\text{SR})_{26}]^{2-}$ (right) NCs. The resolving gel was prepared by 30 w.t.% of acrylamide monomer with a running voltage and time of 160 V and 2.5 h, respectively. The band fronts are indicated by dotted lines for easy comparison.

10) p. 3 I am not sure that a discrepancy between experimental and simulated mass spectra should be attributed to LOW signal-to-noise (and it is not clear what “better defined” means to me). However, since this is not discussed in the actual text, it is up to the other reviewer if this point was reasonably explained.

Reply: The background noise could interfere the mass spectrum in terms of peak position and peak width, which is especially true at a low signal-to-noise ratio. That is the reason we have seen relatively distinct (but < 0.5 Da) deviation of experimental and simulation spectra for the low signal-to-noise peaks. Within such reasonable deviation range, we present isotope patterns closest to the simulated ones (of which “better defined” is referring to) in the last revision of Supplementary Information. We would like to point out that, with < 0.5 Da deviation, the mass spectra reported in this manuscript are among the best-matched ones ever documented.

11) p. 12 Since the spectra are not normalized, this should be stated in the methods section. Were the ESI-MS and UV-vis measurements performed on the same sample for the various times? (see p. 9 of the main text) If so, how could that affect/not affect the results?

Reply: Yes, the time-dependent UV-vis and ESI-MS analyses were carried out on the same sample for experimental consistency. The UV-vis absorption spectra were not normalized at specific wavelengths after measurement. However, as a common practice to minimize any background interference, we “auto-zero” the absorbance at the edge of our wavelength window (i.e., 1100 nm) and made reference to blank solvent (i.e., water) in individual tests. The ESI-MS intensity was normalized to the total ion count from each measurement. We have added aforementioned characterization details in the section Methods to eliminate any possible ambiguity.

Revisions:

Page 29, Lines 1 – 2:

“The absorbance of reaction mixture was reset (to zero) and made reference to that of ultrapure water in individual test.”

Page 29, Lines 10 – 11:

“The time-dependent ESI-MS spectra were normalized to the total ion count in individual test.”

Summary:

Overall, the authors have done a nice job of elucidating some of the intermediates that can be involved in nanoparticle growth. The mechanisms are still a bit uncertain, but it will be a while before these can be determined one way or another. It is good that the authors have now phrased things more as suggestions rather than as proven mechanisms. The paper will appeal to specialists in this area, but I am not sure it will appeal to the broader Nature Communications audience. A more specialized journal may still be more appropriate.

Reply: We are greatly encouraged by the positive acknowledgement of the reviewer on significance, scientific content and improved readability of our manuscript. We are also in complete agreement with the reviewer’s view that even more in-depth studies should be carried out in the near future to uncover the particle growth mechanisms unambiguously and precisely at the more advanced atomic-level. In this present revision, we believe that the above articulations have convinced the reviewer that the current work on molecular-level growth mechanism (i.e., a complete set of step reactions) represents a necessary and significant step towards such many others’ ultimate goal (i.e., atomic-level reaction dynamics in each step reaction); is beneficial in general to nanomaterials scientists rather than narrowly to the metal cluster research community. In addition, this work demonstrated a facile preparation method of metal NCs with atomically customizable sizes, which could add to many applicable explorations (e.g., catalysis, healthcare, energy conversion and storage, and environmental monitoring) based on metal NCs. We thus believe this work will be of broad interest to heterogeneous readers of *Nature Communications*.

REVIEWERS' COMMENTS:

Reviewer #3 (Remarks to the Author):

The authors have satisfactorily addressed my comments.